# Mnemonic construction and representation of temporal structure in the hippocampal formation

Jacob L. S. Bellmund [1✉], Lorena Deuker[2], Nicole D. Montijn [3] & Christian F. Doeller [1,4,5✉]

The hippocampal-entorhinal region supports memory for episodic details, such as temporal relations of sequential events, and mnemonic constructions combining experiences for inferential reasoning. However, it is unclear whether hippocampal event memories reflect temporal relations derived from mnemonic constructions, event order, or elapsing time, and whether these sequence representations generalize temporal relations across similar sequences. Here, participants mnemonically constructed times of events from multiple sequences using infrequent cues and their experience of passing time. After learning, event representations in the anterior hippocampus reflected temporal relations based on constructed times. Temporal relations were generalized across sequences, revealing distinct representational formats for events from the same or different sequences. Structural knowledge about time patterns, abstracted from different sequences, biased the construction of specific event times. These findings demonstrate that mnemonic construction and the generalization of relational knowledge combine in the hippocampus, consistent with the simulation of scenarios from episodic details and structural knowledge.

---

[1] Max Planck Institute for Human Cognitive and Brain Sciences, Leipzig, Germany. [2] Donders Institute for Brain, Cognition and Behaviour, Radboud University, Nijmegen, The Netherlands. [3] Department of Clinical Psychology, Utrecht University, Utrecht, The Netherlands. [4] Kavli Institute for Systems Neuroscience, Centre for Neural Computation, The Egil and Pauline Braathen and Fred Kavli Centre for Cortical Microcircuits, Jebsen Centre for Alzheimer's Disease, Norwegian University of Science and Technology, Trondheim, Norway. [5] Wilhelm Wundt Institute of Psychology, Leipzig University, Leipzig, Germany. ✉email: bellmund@cbs.mpg.de; doeller@cbs.mpg.de

Our memories are not veridical records, but constructions of our past[1]. When constructing scenarios of the past or future, we often combine specific episodic details with general, semantic knowledge[2–7]. For example, we can infer the time when an event took place not only from episodic details but also from associative or contextual information and general knowledge[8,9]. To answer the question when you left for work yesterday, you may combine knowledge about usually departing from home around 8:30 a.m. with the specific sequence of events that unfolded – eating breakfast while listening to the 8 a.m. news and arriving at work a few minutes late for the 9 a.m. meeting despite good traffic conditions on your commute. You infer that you left later than usual, at around 8:40 a.m. Thus, constructive mnemonic processes allow you to estimate when this event occurred, even if a specific event time is not part of the original memory[8,9]. Event representations in the hippocampal-entorhinal region carry information about sequence relationships[10,11], but whether this goes back to mnemonic construction is unclear. Next to its role in memory for specific sequences, the hippocampal-entorhinal region also generalizes across experiences via the abstraction of structural regularities and the recombination of information across episodes[12,13], suggesting you may use knowledge about comparable mornings to recall your departure time. Here, we ask whether temporal event relations are generalized across sequences that share a similar structure and address the question how mnemonic construction and generalization combine in the hippocampus and in participants' memory for event times.

In line with its well-established role in episodic memory, the hippocampal-entorhinal region is centrally involved in processing and remembering specific event sequences[10]. For instance, learning sequences recruits the hippocampus and entorhinal cortex[14,15], and hippocampal activity increases at event boundaries delineating sequences[16,17]. Hippocampal multi-voxel patterns are sensitive to objects shown at learned sequence positions[18], and recent work suggests that the hippocampus incorporates the duration of intervals between elements in sequence representations[19,20]. Further, pattern correlations in the hippocampus and entorhinal cortex relate to memory for temporal relations[21–26].

Hippocampal and entorhinal representations of events occurring in sequence reflect the temporal relations of these events. In one experiment, participants learned the spatial and temporal relationships of events encountered in sequence along a route through a virtual city[21,27]. After relative to before learning, pattern similarity in the anterior hippocampus and the anterior-lateral entorhinal cortex reflected the temporal sequence relationships between pairs of events. Events closer in time elicited more similar activity patterns relative to events separated by longer intervals, resulting in negative correlations between pattern similarity and temporal distances[21,27]. Within the entorhinal cortex, this effect was specific to the anterior-lateral subregion[27], consistent with the involvement of this area in precise temporal memory recall[28,29]. Negative correlations between pattern similarity and distances are in line with sequence representations akin to cognitive maps of space – positions separated by low distances share similar representations, whereas positions with high distances between them are represented less similarly, i.e. pattern similarity scales with distance.

However, whether event representations in the anterior hippocampus and anterior-lateral entorhinal cortex reflect temporal distances based on constructed event times is unclear. Alternatively, these representations of temporal structure could relate to the order of events. For example, successive events could be linked together, resulting in representations of sequence order, where temporal distances are defined based on the number of

associative links between events[30–32]. Another possibility is that temporal structure representations arise through elapsing time more passively. For example, the firing of individual entorhinal neurons changes with varying time constants in rodents and non-human primates, allowing time to be decoded from population activity[33,34]. Slowly drifting activity patterns could be incorporated into event memories as temporal tags, providing a potential mechanism for temporal memory[35]. Here, we tested whether event representations reflect temporal relations based on mnemonically constructed event times, even when accounting for event order and objectively elapsing time.

Mnemonic construction enables prospective cognition[2,5,36]. The hippocampal-entorhinal region integrates and recombines episodic details across experiences for future simulation, inferential reasoning and generalization[5,12,13,37–40]. Work in rodents and humans demonstrates that the hippocampus supports transitive inference, which requires inferring novel relations between stimulus pairs from knowledge about previously learned premise pairs[41–43]. Further, it combines separately learned associations, enabling inferences about shared associations[44–50]. Recent work suggests a central role for the entorhinal cortex in the abstraction of structural knowledge that is linked to sensory experience in the hippocampus[12,51]. Indeed, entorhinal activity patterns reflected structural similarities between choice options in a reinforcement learning task[52]. Furthermore, in an associative inference task, hippocampal activity patterns carried information about the shared internal structure of image triads such that the hippocampal representational geometry was generalized across triads[53]. Work in rodents suggests that hippocampal representations of events in a sequence generalize across comparable experiences in a different environment[54]. Applying abstract structural knowledge enables adaptive behavior through the generalization of relations to novel situations[12,51]. Whether representations of temporal relations of events in a sequence are constructed such that they generalize across sequences with a similar structure is unclear.

Knowledge about structural regularities and semantic associations closely interacts with episodic construction[4,7,38]. When estimating the size of studied images, participants' reconstructions were systematically distorted towards category averages[55,56]. For relatively small fruits like strawberries, participants tended to overestimate the studied size, whereas they consistently underestimated sizes of large fruits like pineapples. This resulted in an overall bias towards the category mean of all fruits[55]. Consistent with the notion that learned event structures contribute to event cognition[57–59], external and semantic details are used to furnish past and future scenarios when few episodic details are generated[60,61]. When estimating the times of events from a movie, which was terminated prematurely, participants underestimated when events took place for events close to the end of the presented section, possibly due to prior knowledge about the typical structure of movie plots[62]. These findings suggest that abstract knowledge about general patterns could systematically distort constructions of specific event times. If, as in the introductory example, you usually leave for work at 8:30 a.m., this may bias the estimate of your departure time on the day you arrived late towards this time.

Here, we combine functional magnetic resonance imaging (fMRI) with a sequence learning task requiring the memory-based construction of the times of events forming different sequences. We show that event representations in the anterior hippocampus change through learning to reflect constructed event times rather than sequence order or passively elapsing time. Furthermore, the anterior hippocampus generalizes temporal relations across sequences, and structural knowledge about other sequences systematically biases the construction of specific event times. While within- and across-sequence relations are detected in anatomically

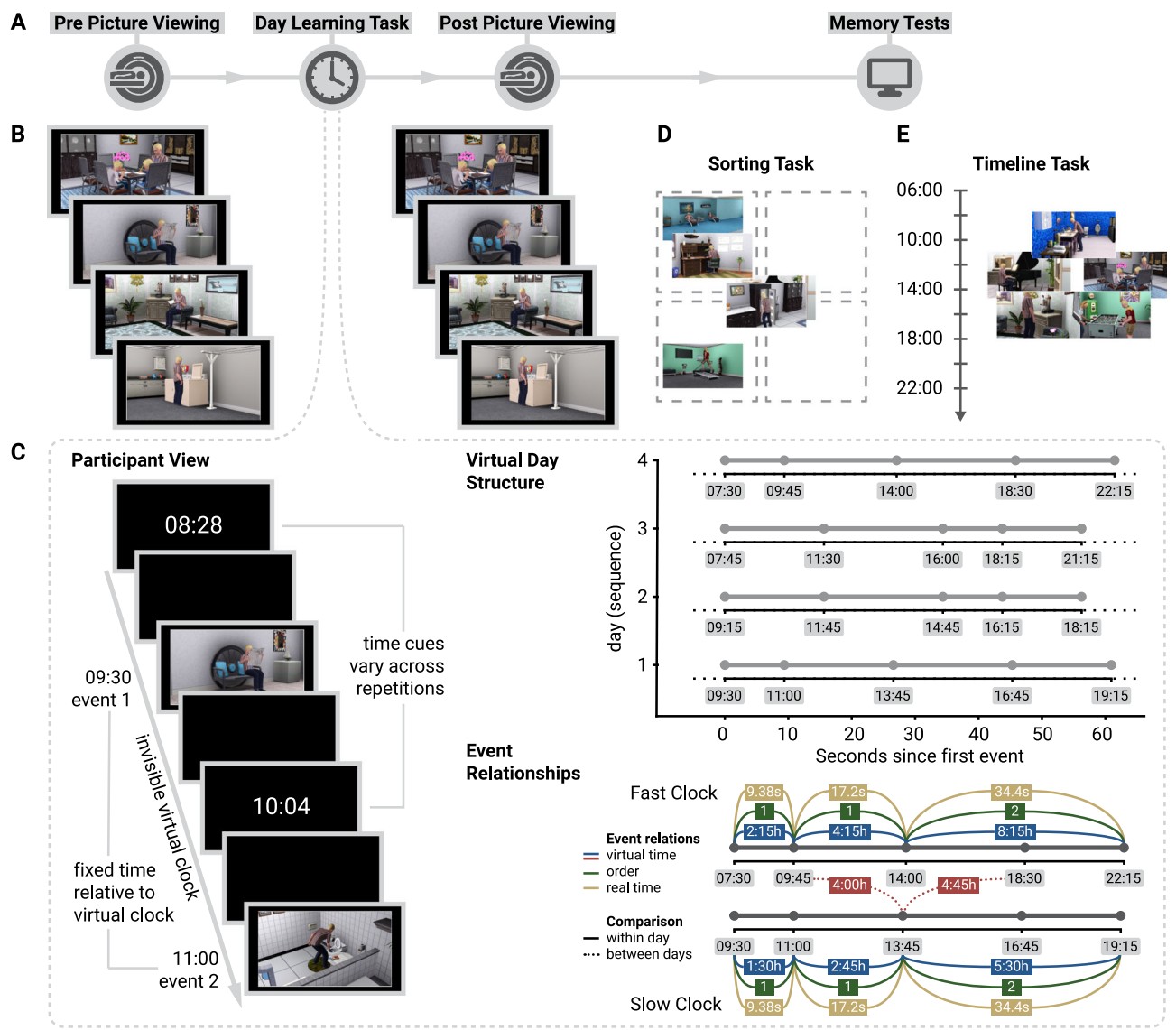

**Fig. 1 Experimental design. A** Overview of the experiment. **B** In the picture viewing tasks before and after learning, participants saw event images presented in the same random order and using identical stimulus timings. **C** The day learning task took place in between the picture viewing tasks. Participants learned four sequences (virtual days) of five events each (Supplementary Fig. 1) and inferred when events took place relative to a virtual clock. Left: The virtual clock ran hidden in the background for each sequence and was revealed only once in between successive events. These time cues varied across repetitions of a sequence, but events occurred at consistent points in virtual time. The duration of blank screen periods varied according to the interval between the indicated time and the event time. Thus, participants had to mentally construct event times by combining their experience of elapsing real time with the time cues. Top right: The hidden clock ran at a fixed speed relative to real time for a given sequence, but its speed varied between sequences (Supplementary Fig. 2). Bottom right: Different time metrics capture the temporal structure of the event sequences. Event relations can be quantified using temporal distances relative to the hidden clock (virtual time), sequence positions (order), and elapsed time in seconds (real time). While these metrics inevitably covary, they are partially dissociated by the clock speed manipulation. Virtual temporal distances can be quantified both within (solid lines) and across sequences (dotted lines). **D**, **E** Participants' memory of the sequences was tested in two tasks. In the sorting task (**D**), participants sorted the scenes according to the four different sequences. In the timeline task (**E**), participants positioned the five event images of a given sequence next to a timeline to indicate constructed event times. **B**–**E** The Sims 3 and screenshots of it are licensed property of Electronic Arts, Inc. All rights are reserved.

overlapping regions of the hippocampus, the mode of representation differs depending on whether events belong to the same sequence or not. In contrast, the anterior-lateral entorhinal cortex uses one shared representational format to map relationships of events from the same and from different sequences.

## Results

We asked participants to learn four sequences that consisted of five unique event images each (Fig. 1). Participants were

instructed that each sequence depicted events taking place on a specific day in the life of a family. Their task was to infer the time of each event relative to the temporal reference frame of a virtual clock (Fig. 1C). Event images with minimal or no indication of time of day (Supplementary Fig. 1) were randomly assigned to sequences and sequence positions for each participant. Thus, it was impossible to infer specific event times or sequence memberships from the stimuli. The true virtual times of events were never revealed. Rather, the clock was running hidden from participants. It was uncovered only infrequently between event

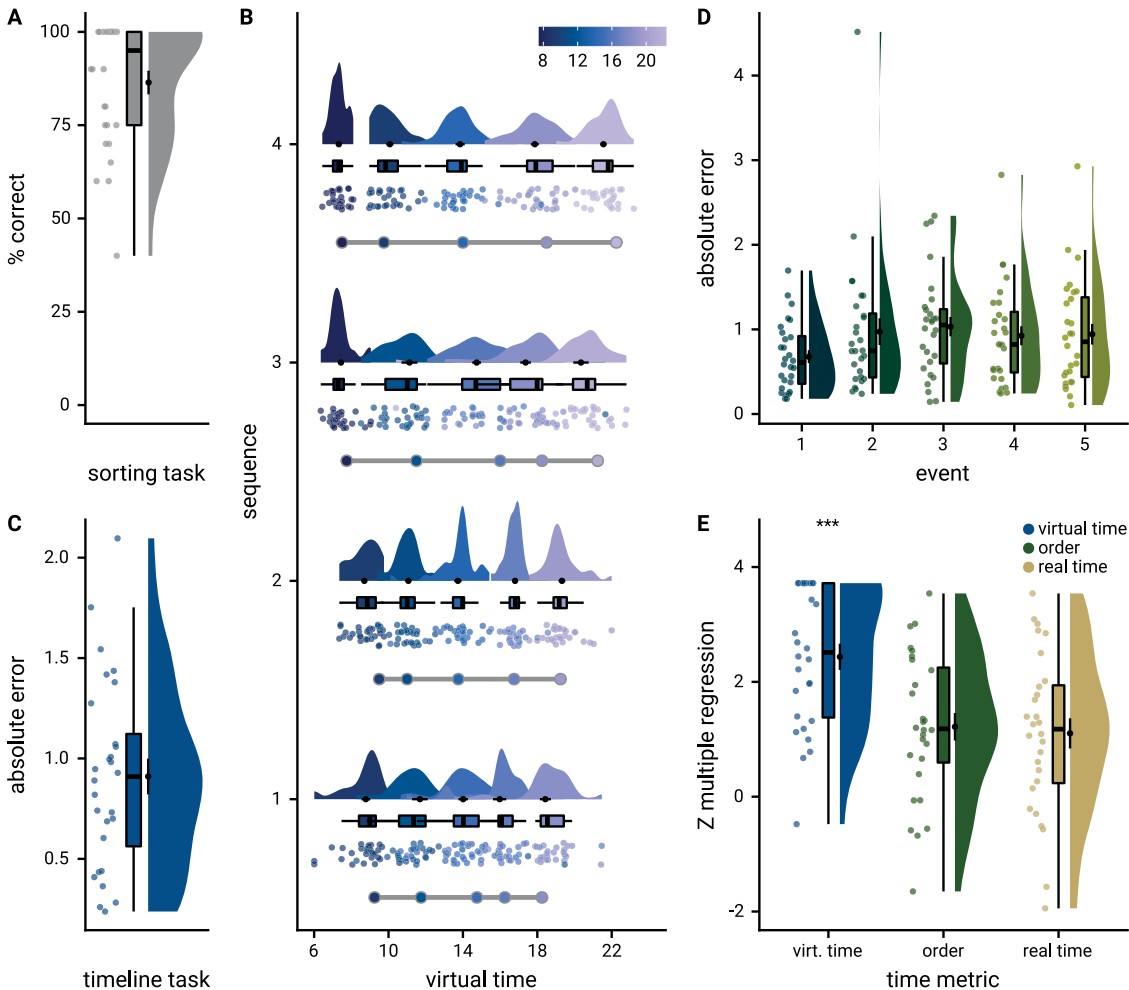

**Fig. 2 Participants learn the temporal structure of the sequences relative to the virtual clock. A** Plot shows the percentage of correctly sorted event images in the sorting task. **B** Constructed event times were assessed in the timeline task. Responses are shown separately for the five events (color coded according to true virtual time) of each sequence (rows). Colored circles with gray outline show true event times. Mean absolute errors in constructed times (in virtual hours) are shown (**C**) averaged across events and sequences and (**D**) averaged separately for the five event positions. **E** Z-values for the effects of different time metrics from participant-specific multiple regression analyses and permutation tests show that virtual time explained constructed event times with event order and real time in the model as control predictors. **A**-**E** Circles show data from $n = 28$ participants; boxplots show median and upper/ lower quartile along with whiskers extending to most extreme data point within 1.5 interquartile ranges above/below the upper/lower quartile; black circle with error bars corresponds to mean ± S.E.M.; distributions show probability density function of data points. Source data are provided as a Source Data file. ***$p < 0.001$.

presentations to briefly show the current virtual time (Supplementary Fig. 2, see Methods). Participants had to combine their experience of objectively elapsing time (real time) with the virtual time cues to construct event times. Importantly, we manipulated the speed of the hidden clock between sequences so that different amounts of virtual time passed in the same real time intervals. With this paradigm, we partially dissociated the virtual time of events from the event order and real time to test whether mnemonically constructed event times underlie participants' memory for the temporal structure of the sequences.

**Successful construction of event times**. We assessed memory for the sequences using two behavioral tests administered at the end of the experimental session. First, participants sorted all event images according to sequence membership (Fig. 1D). The high performance in this task (Fig. 2A; 86.43% ± 16.82% mean ± standard deviation of correct sorts) demonstrates accurate memory for which events belonged to the same sequence. The distribution of sorting errors did not differ from uniformity

across sequence positions ($\chi^2 = 2.55$, p = 0.635). Second, to probe constructed event times, we asked participants to position the events of a sequence on a timeline (Fig. 1E). Remembered times were highly accurate (Fig. 2B–D; 0.91 ± 0.47 mean ± standard deviation of average absolute errors in virtual hours). The accuracy of constructed virtual times differed between sequences ($F_{3,81} = 5.86$, $p < 0.001$), but not as a function of virtual clock speed ($t_{27} = -0.82$, $p = 0.423$, Supplementary Fig. 3A, B). We did not observe an across-subject relationship between the number of sorting errors and mean absolute errors in the timeline task (Supplementary Fig. 3C, D). To test whether the constructed event times were driven by the virtual time of events, we regressed remembered times on virtual times with event order and real time as control predictors of no interest. We did so in a summary statistics approach based on multiple regression for each participant, combined with permutation tests, and using a linear mixed effects model (see Methods). The effect of virtual time on constructed event times was significant when controlling for variance accounted for by event order and real time (Fig. 2E; summary statistics: $t_{27} = 10.62$, $p < 0.001$, $d = 1.95$, 95% CI [1.38,

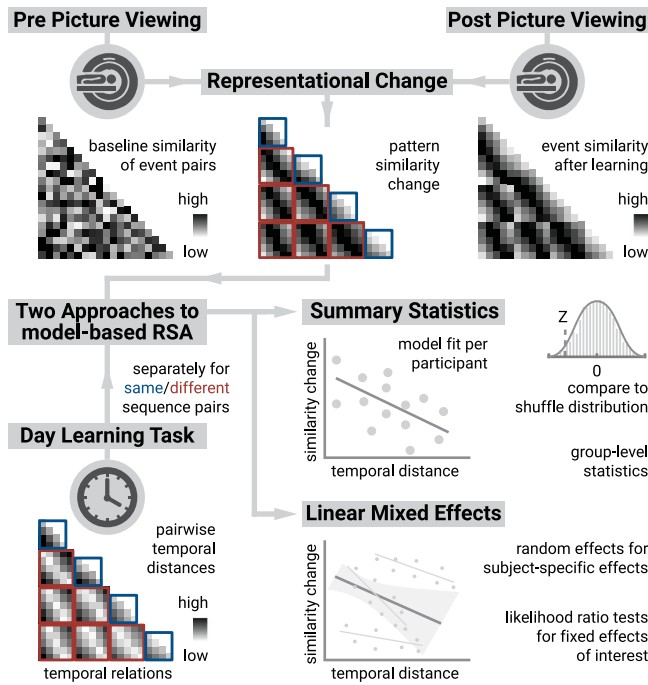

**Fig. 3 Representational similarity analysis logic.** We quantified the representational similarity of all event pairs before and after learning. Representational change was defined by subtracting pre-learning from post-learning pattern similarity (top row). Using two approaches to model-based representational similarity analysis (RSA, see Methods), we analyzed whether pattern similarity changes reflected the temporal structure of the sequences (bottom left). In the summary statistics approach (middle right), we regressed pattern similarity change on temporal distances between events using participant-specific linear models that were compared to null distributions obtained from shuffling similarity change against temporal distances. The resulting Z-values were used for permutation-based group-level statistics. In the mixed model approach (bottom right), we estimated the influence of temporal distances on pattern similarity change using fixed effects, with random effects accounting for within-subject dependencies. The statistical significance of fixed effects was assessed using likelihood ratio tests against reduced models excluding the fixed effect of interest.

2.70]; mixed model: $\chi^2(1) = 115.95$, $p < 0.001$, Supplementary Fig. 4A, B, Supplementary Table 1). Together, these findings demonstrate that participants formed precise memories of the different sequences and accurately constructed event times.

**Hippocampal representations of within-sequence relations reflect constructed event times.** Before and after learning the event sequences, participants viewed the event images in random order while undergoing fMRI (Fig. 1A, B). We quantified changes in the similarity of multi-voxel patterns between pairs of events from before to after learning (Fig. 3, see Methods). Using two statistical approaches for model-based representational similarity analysis, we tested whether changes in pattern similarity could be explained by the temporal relationships between pairs of events. Temporal distances between events were measured in virtual time, real elapsing time in seconds and as differences in sequence order position (Fig. 1C). In the summary statistics approach, we compared the fit of linear models predicting pattern similarity changes from temporal distances to shuffle distributions for each participant and assessed the resulting Z-values on the group level using permutation-based tests. Second, we fitted linear mixed effects models to quantify whether sequence relationships explained pattern similarity changes. Rather than performing inferential

statistics on one summary statistic per participant, mixed models estimate fixed effects and their interactions using all data points. We used temporal distance measures as fixed effects while capturing within-participant dependencies with random intercepts and random slopes (see Methods). The converging results of these analyses demonstrate that our findings do not depend on the specific statistical methods employed. We centered our analyses on the anterior hippocampus and the anterior-lateral entorhinal cortex (see Methods) based on our previous work implicating these regions in representing sequence relations[21,27].

We first tested whether pattern similarity changes in the anterior hippocampus (Fig. 4A) could be explained by the virtual temporal distances between event pairs from the same sequence. Surprisingly, we observed a positive relationship between similarity changes and temporal distances in both the summary statistics (Fig. 4B; $t_{27} = 3.07$, $p = 0.006$, $d = 0.56$, 95% CI [0.18, 1.00]; $\alpha = 0.025$, corrected for separate tests of events of the same and different sequences) and the mixed model approach (Fig. 4C, D; $\chi^2(1) = 9.87$, $p = 0.002$, Supplementary Fig. 4C, D, Supplementary Table 2). This effect was further characterized by higher pattern similarity for event pairs separated by longer temporal distances than for pairs separated by shorter intervals (Fig. 4C, $t_{27} = 2.48$, $p = 0.020$, d = 0.64, 95% CI [0.08, 0.87]). In contrast to our previous work[21], where we observed negative correlations of pattern similarity and temporal distances, participants learned multiple sequences in this study. They might have formed strong associations of same-sequence events on top of inferring each event's virtual time, potentially altering how temporal distances affected hippocampal pattern similarity (see Discussion). The effect of virtual temporal distances on pattern similarity changes remained significant when competing for variance with a control predictor accounting for comparisons of the first and last event of each sequence (Supplementary Fig. 5A–C; summary statistics: $t_{27} = 2.25$, $p = 0.034$, $d = 0.41$, 95% CI [0.04, 0.82]; mixed model: $\chi^2(1) = 5.36$, $p = 0.021$, Supplementary Table 3). Thus, the relationship of hippocampal event representations and temporal distances is not exclusively driven by associations of the events marking the transitions between sequences.

Having established that hippocampal pattern similarity changes relate to temporal distances, we next assessed whether this effect was driven by virtual event times beyond sequence order and real time. We thus included the two additional time metrics as control predictors in the model. Virtual temporal distances significantly predicted pattern similarity changes even when controlling for the effects of event order and real time in seconds (Fig. 4D; summary statistics: $t_{27} = 2.18$, $p = 0.040$, $d = 0.40$, 95% CI [0.02, 0.81]; mixed model: $\chi^2(1) = 5.92$, $p = 0.015$, Supplementary Fig. 4E, F, Supplementary Table 4). Further, the residuals of linear models, in which hippocampal representational change was predicted from order and real time, were related to virtual temporal distances (Supplementary Fig. 5D; $t_{27} = 2.23$, $p = 0.034$, $d = 0.41$, 95% CI [0.03, 0.82]), demonstrating that virtual time accounts for variance that the other time metrics fail to explain. Together, these data show that hippocampal representations of events from the same sequence changed to reflect mnemonically constructed event times.

**The hippocampus generalizes temporal relations across sequences.** We next tested whether similarity changes of hippocampal representations of events from different sequences mirrored generalized temporal distances. When comparing pairs of events belonging to different sequences, we observed a significant negative effect of virtual temporal distances on pattern similarity change (Fig. 5A, summary statistics $t_{27} = -2.65$, $p = 0.013$, $d = -0.49$, 95% CI [−0.91, −0.10]; mixed model: $\chi^2(1) = 6.01$,

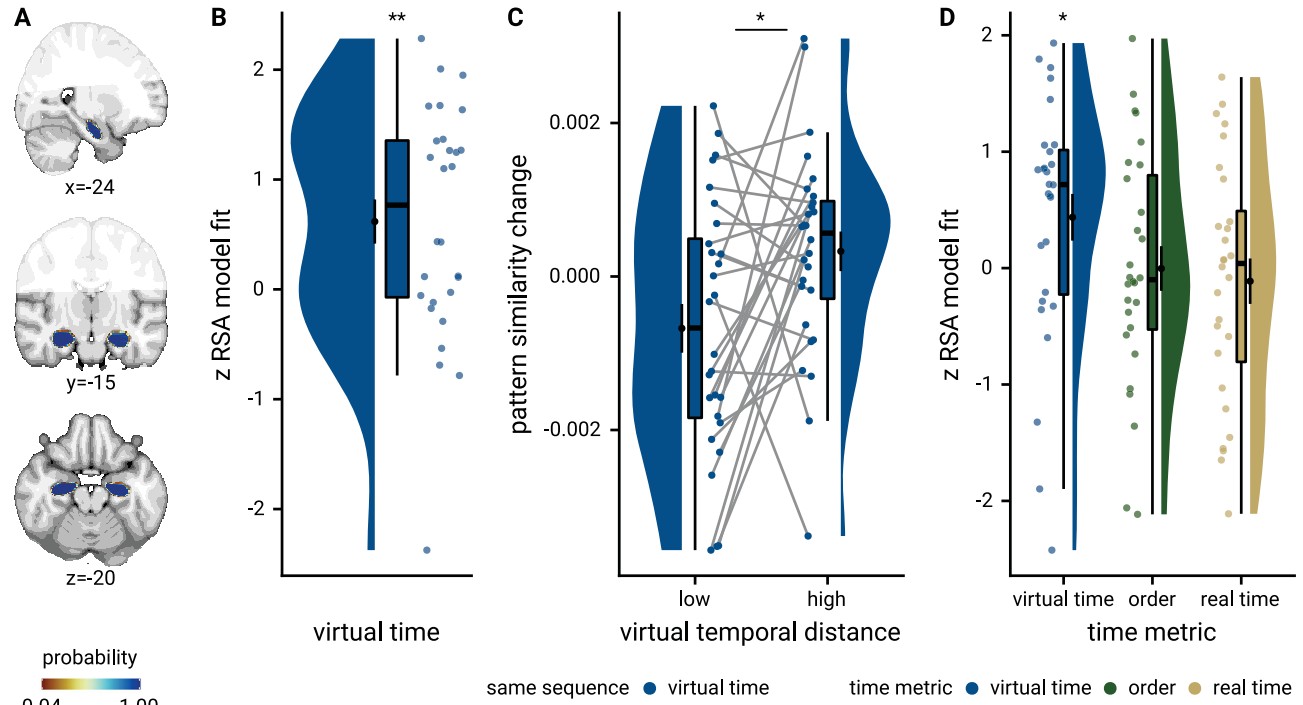

**Fig. 4 Sequence representations in anterior hippocampus reflect constructed event times. A** The anterior hippocampus region of interest is displayed on the MNI template with voxels outside the field of view shown in lighter shades of gray. Color code denotes probability of a voxel to be included in the mask based on participant-specific ROIs (see Methods). **B** The Z-values based on permutation tests of participant-specific linear models assessing the effect of virtual time on pattern similarity change for event pairs from the same sequence were significantly positive. **C** To illustrate the effect shown in **B**, average pattern similarity change values are shown for same-sequence event pairs that are separated by low and high temporal distances based on a median split. **D** Z-values show the relationship of the different time metrics to representational change based on participant-specific multiple regression analyses. Virtual time predicts pattern similarity change with event order and real time in the model as control predictors of no interest. **B–D**. Circles show data from $n = 28$ participants; boxplots show median and upper/lower quartile along with whiskers extending to most extreme data point within 1.5 interquartile ranges above/below the upper/lower quartile; black circle with error bars corresponds to mean ± S.E.M.; distributions show probability density function of data points. Source data are provided as a Source Data file. **\*\***$p < 0.01$; **\***$p < 0.05$.

$p = 0.014$, Supplementary Fig. 4G, H, Supplementary Table 5; $\alpha = 0.025$, corrected for separate tests of events of the same and different sequences). This indicates that hippocampal representations of events from different sequences changed systematically to reflect generalized temporal relations. Events occurring at similar times relative to the virtual clock, but in different sequences, were represented more similarly than those taking place at more different virtual times (Fig. 5B, $t_{27} = -3.26$, $p = 0.002$, $d = -0.89$, 95% CI [$-1.03$, $-0.21$]). Virtual time was a significant predictor of hippocampal pattern similarity change for events from different sequences when competing for variance with order and real time (Supplementary Fig. 6A–C; summary statistics: $t_{26} = -2.62$, $p = 0.015$, $d = -0.49$, 95% CI [$-0.92$, $-0.10$], mixed model: $\chi^2(1) = 4.48$, $p = 0.034$, Supplementary Table 6; one outlier excluded). The relationship of temporal distances and representational change differed significantly between events from the same or different sequences (Fig. 5A, summary statistics: paired $t$-test $t_{27} = 3.71$, $p = 0.001$, $d = 1.05$, 95% CI [0.29, 1.13]; mixed model: interaction of sequence membership with virtual time $\chi^2(1) = 14.37$, $p < 0.001$, Supplementary Fig. 4I, J, Supplementary Table 7). Similar interactions of sequence membership with order ($\chi^2(1) = 9.98$, $p = 0.002$) and real time ($\chi^2(1) = 9.27$, $p = 0.002$) were observed, but, crucially, the interaction of sequence membership and virtual time remained significant when including interactions of sequence membership with order and real time in the model ($\chi^2(1) = 8.57$, $p = 0.003$, Supplementary Table 8). Thus, the way knowledge about virtual temporal relations was represented in the

hippocampus depended on whether events belonged to the same sequence or not.

To explore how event sequences may be arranged in a low-dimensional representational space to give rise to the effects described above, we generated a distance matrix from the mixed effects model fitted to hippocampal pattern similarity change and subjected it to non-metric multidimensional scaling (see Methods, Supplementary Fig. 6D). The resulting configuration in two dimensions (Fig. 5C), chosen for intuitive visualization, exhibited a c-shaped pattern for each sequence. Similar representational geometries have previously been described in parietal cortex[63–65]. Events occurring at similar virtual times occupy similar locations, in line with high pattern similarity for events from different sequences that are separated by low temporal distances. Thus, the generalization across sequences results in a comparable configuration for each sequence. While the observed configuration resulted in stress values significantly lower than those obtained in a permutation test (see Methods; $z = -3.5$, $p = 0.001$, Supplementary Fig. 6E), the high representational distances between temporally close events from the same sequence are not perfectly captured by the c-shaped arrangement (Supplementary Fig. 6F, G). More than the two dimensions chosen for visualization would likely better capture the complex representational structure of the sequences.

**Sequence representations differ between hippocampus and entorhinal cortex.** In our second region of interest, the anterior-lateral entorhinal cortex (Fig. 6A), the effect of virtual time on

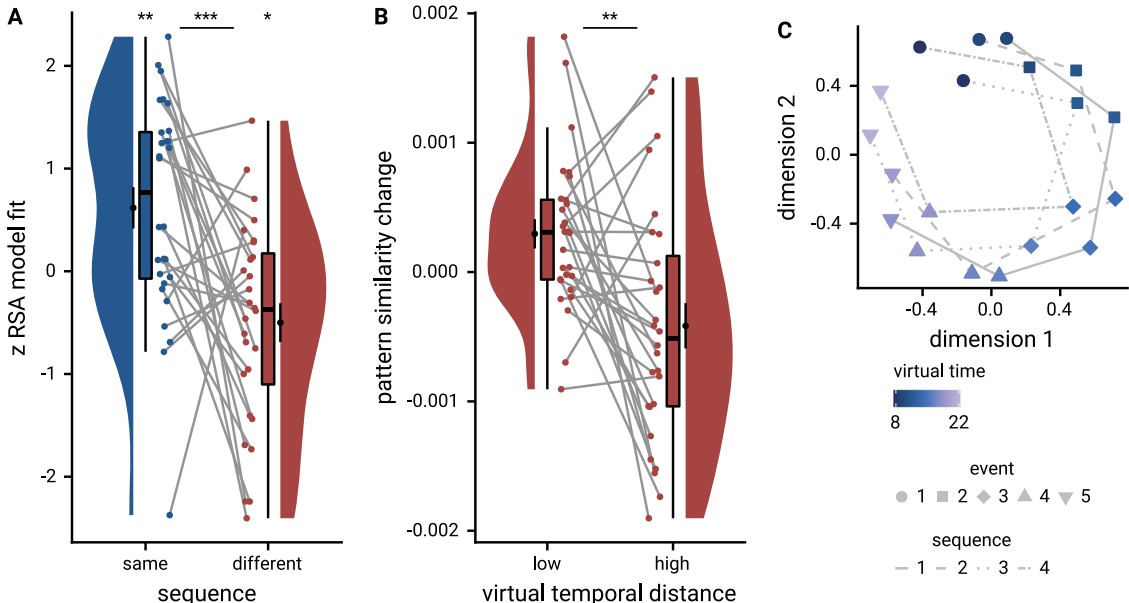

**Fig. 5 The anterior hippocampus generalizes temporal relations across sequences. A** Z-values show results of participant-specific linear models quantifying the effect of virtual time for event pairs from the same sequence (blue, as in Fig. 4B) and from different sequences (red). Temporal distance is negatively related to hippocampal representational change for event pairs from different sequences. See Supplementary Fig. 4E, F for mixed model analysis of across-sequence comparisons. The effect of virtual time differs for comparisons within the same sequence or between two different sequences. **B** To illustrate the effect shown in **A**, average pattern similarity change values are shown for across-sequence event pairs that are separated by low and high temporal distances based on a median split. **A, B**. Circles show data from n = 28 participants; boxplots show median and upper/lower quartile along with whiskers extending to most extreme data point within 1.5 interquartile ranges above/below the upper/lower quartile; black circle with error bars corresponds to mean ± S.E.M.; distributions show probability density function of data points. **p < 0.001; **p < 0.01; *p < 0.05 **C** Multidimensional scaling results show low-dimensional embedding of the event sequences. Shapes indicate event order, color shows virtual times of events. The different lines connect the events belonging to the four sequences for illustration. Source data are provided as a Source Data file. ***p ≤ 0.001; **p < 0.01; *p < 0.05.

representational change did not differ statistically between event pairs from the same or from different sequences (summary statistics: paired *t*-test $t_{27} = 0.07$, $p = 0.942$). We thus collapsed across comparisons from the same and different sequences and observed a significant effect of virtual temporal distances on entorhinal pattern similarity change (Fig. 6B; summary statistics: $t_{27} = -2.31$, $p = 0.029$, $d = -0.42$, 95% CI [−0.84, −0.05]; mixed model: $\chi^2(1) = 4.39$, $p = 0.036$, Supplementary Fig. 4K, L, Supplementary Table 9; see Supplementary Fig. 7A for separate analyses of events from the same and from different sequences). In line with our previous work[27], events close together in time became more similar than those separated by longer temporal intervals (Fig. 6C). The relationship of virtual temporal distances and entorhinal pattern similarity change was not statistically significant when competing for variance with distances based on order and real time (Supplementary Fig. 7B–D; summary statistics: $t_{27} = -0.7$, $p = 0.495$, $d = -0.13$, 95% CI [−0.51, 0.25], mixed model: $\chi^2(1) = 1.18$, $p = 0.278$, Supplementary Table 10). We further corroborated that the temporal structure of the sequences was represented differently between the anterior-lateral entorhinal cortex and the anterior hippocampus (summary statistics: interaction between region and sequence membership in permutation-based repeated-measures ANOVA $F_{1,27} = 7.76$, $p = 0.010$, $\eta^2 = 0.08$, main effect of region $F_{1,27} = 3.10$, $p = 0.086$, $\eta^2 = 0.02$, main effect of sequence $F_{1,27} = 7.41$, $p = 0.012$, $\eta^2 = 0.08$; mixed model: three-way interaction between virtual time, sequence membership and region of interest $\chi^2(1) = 6.31$, $p = 0.012$, Supplementary Table 11; see Supplementary Fig. 8 for a comparison of the signal-to-noise ratio in these regions). Whereas the hippocampus employed two distinct representational formats for temporal relations depending on whether events belonged to the same sequence or not, we observed consistent negative correlations between representational change

and temporal distances when collapsing across all event pairs, but no statistically significant difference between representations of temporal relations from the same or different sequences in the entorhinal cortex.

**Anatomical overlap between representations of within-sequence relations and across-sequence generalization.** We next asked whether representations of same-sequence relations are distinct from or overlap with the across-sequence generalization of temporal relations. For this purpose and to complement our region-of-interest analyses described above, we performed a searchlight analysis that revealed significant effects of virtual temporal distances on representations of events from the same sequence in the bilateral anterior hippocampus (Fig. 7A; peak voxel MNI $x = -24$, $y = -13$, $z = -20$; $t = 4.53$, $p_{svc} = 0.006$, Supplementary Table 12). We used the same-sequence searchlight peak cluster to define a region of interest to test for the independent across-sequence generalization effect (see Methods). Indeed, virtual temporal distances explained pattern similarity change for events from different sequences in these voxels (Fig. 7B; summary statistics $t_{27} = -2.19$, $p = 0.036$, $d = -0.40$, 95% CI [−0.81, −0.03]; mixed model: $\chi^2(1) = 4.13$, $p = 0.042$, Supplementary Fig. 4M, N, Supplementary Table 13), demonstrating an overlap between representations of within-sequence relations and their generalization across sequences.

Further, we conducted a searchlight analysis looking for negative correlations of temporal distances and pattern similarity change for events from different sequences. We detected clusters in anterior hippocampus that overlapped with the same-sequence searchlight effect (Fig. 7C, D), though this searchlight generalization effect did not survive corrections for multiple

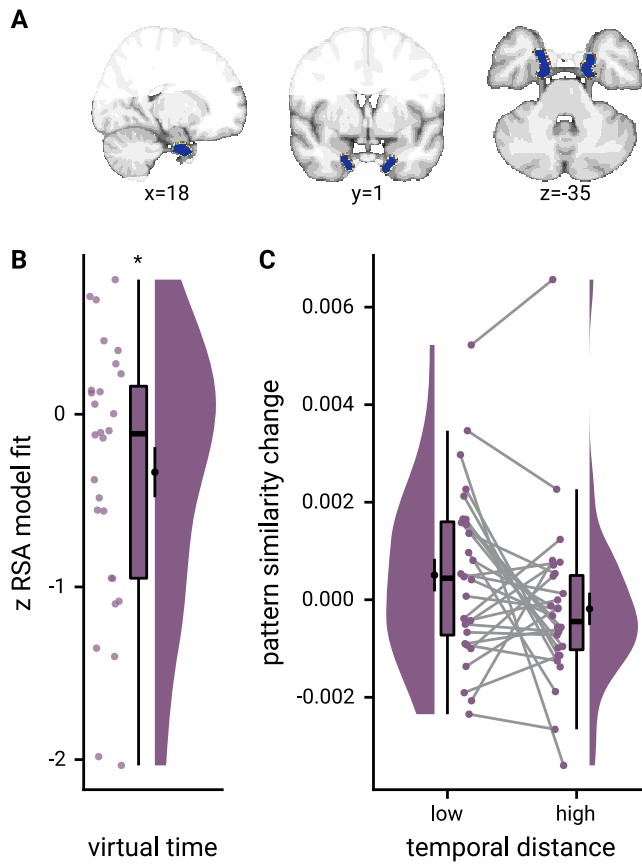

**Fig. 6 The anterior-lateral entorhinal cortex uses a shared representational format for relations of events from the same and different sequences. A** The anterior-lateral entorhinal cortex region of interest is displayed on the MNI template with voxels outside the field of view shown in lighter shades of gray. Color code denotes probability of a voxel to be included based on participant-specific masks (see Methods). **B** Z-values for participant-specific RSA model fits show a negative relationship between pattern similarity change and virtual temporal distances when collapsing across all event pairs. **C** To illustrate the effect in **B**, raw pattern similarity change in the anterior-lateral entorhinal cortex was averaged for events separated by low and high temporal distances based on a median split. **B, C** Circles show data from $n = 28$ participants; boxplots show median and upper/lower quartile along with whiskers extending to most extreme data point within 1.5 interquartile ranges above/below the upper/lower quartile; black circle with error bars corresponds to mean ± S.E.M.; distributions show probability density function of data points. Source data are provided as a Source Data file. *$p < 0.05$.

comparisons (peak voxel MNI $x = -26$, $y = -19$, $z = -15$, $t = -3.96$, $p_{svc} = 0.071$, Supplementary Table 14). Lastly, we directly searched for brain areas in which pattern similarity change differentially scaled with temporal distances depending on whether events were from the same or different sequences. The two largest clusters in our field of view were located in the left and right anterior hippocampus (Fig. 7E, peak voxel MNI $x = 31$, $y = -16$, $z = -21$; $t = 4.25$, $p_{svc} = 0.007$, Supplementary Table 15). Taken together, these findings highlight that hippocampal representations carry information about the specific sequence in which events occur, and that these temporal relations are generalized across sequences.

**Generalized knowledge about other sequences biases event time construction.** Having established generalized hippocampal event representations, we explored whether knowledge about the

general structure of event times in other sequences influenced the construction of individual event times. For each event, we quantified when it took place relative to the average virtual time of the events at the same sequence position in the other three sequences (Fig. 8A; see Methods). We reasoned that the construction of a specific event time could be biased by knowledge about the general pattern of event times at that sequence position. Indeed, we observed positive relationships between the relative time of other events and signed errors in constructed event times as assessed in the timeline task (Fig. 8B, C, Supplementary Fig. 10A; summary statistics: $t_{27} = 5.32$, $p < 0.001$, $d = 0.98$, 95% CI [0.55, 1.48]; mixed model: $\chi^2(1) = 17.90$, $p < 0.001$, Supplementary Fig. 4O, P, Supplementary Table 16). This demonstrates that structural knowledge about the sequences biased the construction of event times. The constructed virtual time of an event tended to be overestimated when the events occupying the same sequence position in the other sequences took place late relative to the event in question, and vice versa when the other events occurred relatively early. In an independent group of participants[66], we replicated this generalization bias (Fig. 8D, Supplementary Fig. 10B; summary statistics: $t_{45} = 11.30$, $p < 0.001$, $d = 1.64$, 95% CI [1.23, 2.13]; mixed model: $\chi^2(1) = 53.74$, $p < 0.001$, Supplementary Fig. 4Q, R, Supplementary Table 17), confirming the influence of generalized knowledge about the sequences on event time construction. One possibility is that structural knowledge about the sequences biases the construction of specific event times, in particular when uncertainty about the virtual time of events is high. Indeed, we observed a significant negative correlation between how strongly pattern similarity changes in the anterior hippocampus reflected temporal relations between same-sequence events in the searchlight analysis and the strength of the behavioral generalization bias (Fig. 8E, F, Spearman $r = -0.53$, $p = 0.005$; $\alpha = 0.025$ corrected for two comparisons; correlation with across-sequence effect: Spearman $r = -0.19$, $p = 0.322$), suggesting that the construction of event times was less biased by time patterns generalized across sequences in those participants with precise representations of within-sequence temporal relations.

We further explored whether participants made systematic errors in the sorting task that might point towards generalization across sequences. Specifically, we searched for swap errors where participants interchanged events between sequences that occupied the same sequence position. Indeed, 57.5% ± 34.3% (mean ± S.D) of sorting errors were swap errors and 12 of the 14 participants who made sorting errors also made swap errors (Supplementary Fig. 3E, F, mean ± S.D of 3.1 ± 2.1 swap errors per participant with sorting errors). The proportion of swap errors in our sample was larger than expected from random sorting errors ($z = 5.07$, $p < 0.001$, Supplementary Fig. 3G), indicating that participants systematically swapped events belonging to the same position between sequences. While we did not observe statistically significant relationships between swap errors and the generalization bias (Supplementary Fig. 3H, I), the prevalence of these errors is compatible with the view that participants generalized across events occupying the same sequence position.

## Discussion

Our findings show that hippocampal event representations change through learning to reflect temporal relations based on mnemonically constructed event times. Converging region of interest and searchlight analyses demonstrate that, on the one hand, the hippocampus forms specific representations of temporal relations of the events in a sequence that mirror constructed event times beyond the effects of order and real time. On the other hand, temporal relations are generalized across sequences

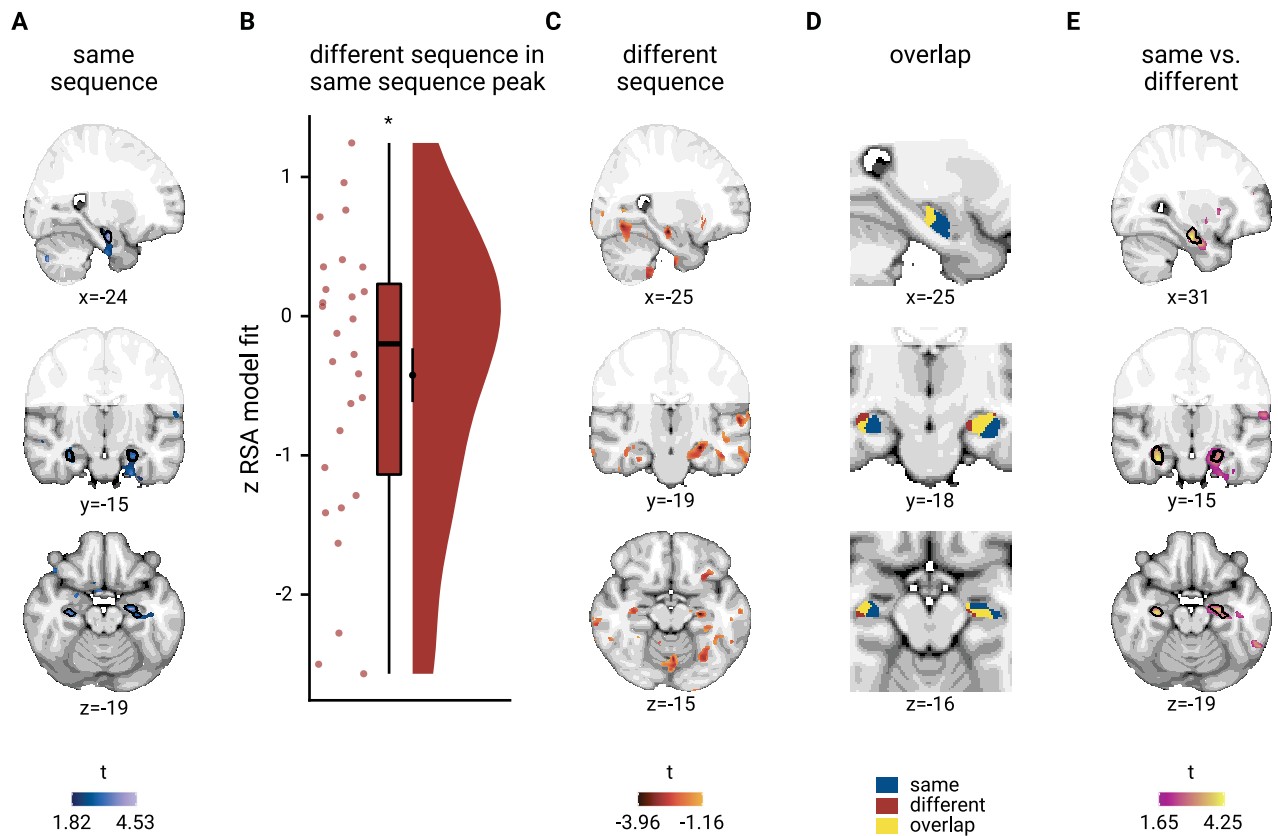

**Fig. 7 Overlapping representations of within- and across-sequence relations. A** Searchlight analysis results show a positive relationship between representational change and virtual temporal distances for event pairs from the same sequence in the bilateral anterior hippocampus. Statistical image is thresholded at $p_{uncorrected} < 0.01$; voxels within black outline are significant after correction for multiple comparisons using small volume correction. **B** In the peak cluster from the independent within-sequence searchlight analysis (**A**), representational change was negatively related to virtual temporal distances between events from different sequences. Circles show Z-values from summary statistics approach for $n = 28$ participants; boxplot shows median and upper/lower quartile along with whiskers extending to most extreme data point within 1.5 interquartile ranges above/below the upper/lower quartile; black circle with error bars corresponds to mean ± S.E.M.; distribution shows probability density function of data points. **C** Searchlight analysis results show negative relationship between representational change and temporal distances for different-sequence event pairs. Statistical image is thresholded at $p_{uncorrected} < 0.05$. **D** Within the anterior hippocampus, the effects for events from the same sequence and from two different sequences overlap. Visualization is based on statistical images thresholded at $p_{uncorrected} < 0.05$ within small volume correction mask. **E** Searchlight analysis results show a bilateral interaction effect in the anterior hippocampus that is defined by a differential relationship of virtual temporal distances and representational change for events from the same and different sequences. Statistical image is thresholded at $p_{uncorrected} < 0.01$; voxels within black outline are significant after correction for multiple comparisons using small volume correction. **A**, **C**–**E** Results are shown on the MNI template with voxels outside the field of view displayed in lighter shades of gray. See Supplementary Fig. 9 for additional exploratory results. Source data are provided as a Source Data file. *$p < 0.05$.

using a different representational format. In contrast, the similarity of event representations in the anterior-lateral entorhinal cortex scaled with temporal distances for events irrespective of sequence membership. The behavioral data demonstrate that the construction of specific event times is biased by structural knowledge abstracted from different sequences.

In our paradigm, participants mentally constructed the times of events relative to a hidden virtual clock. To do so, they needed to combine their experience of passing real time with infrequent cues about the current virtual time. Thus, real time was critical for the successful construction of event times, despite not being cued explicitly. Participants' responses in a memory test and the similarity structure of hippocampal multi-voxel patterns were explained by virtual event times beyond the effects of real time and sequence order, showing that sequence representations reflect mnemonically constructed time. Recent work demonstrated the scaling of time cell representations to different real time intervals in the rodent hippocampus[67]. Temporal scaling of hippocampal

representations could potentially underlie our observation that temporal distances in virtual time are related to the similarity of event representations even when accounting for the effects of real time and order. This finding highlights that the anterior hippocampus maps relational knowledge derived from mnemonic constructions.

The hippocampus constructed an integrated representation that generalized temporal relations across sequences. Multi-voxel patterns of events taking place at similar virtual times, but in different sequences, were more similar than those of events occurring at different points in time. Thus, representations of events from different sequences changed systematically to reflect generalized temporal distances. Speculatively, this effect could be related to the observation that, in mice trained to run a number of laps on a maze to obtain rewards, lap-specific firing patterns in the hippocampus generalize across sequences of laps on geometrically distinct mazes[54]. While it is possible that the first and last events of the sequences are particularly important to sequence processing, our

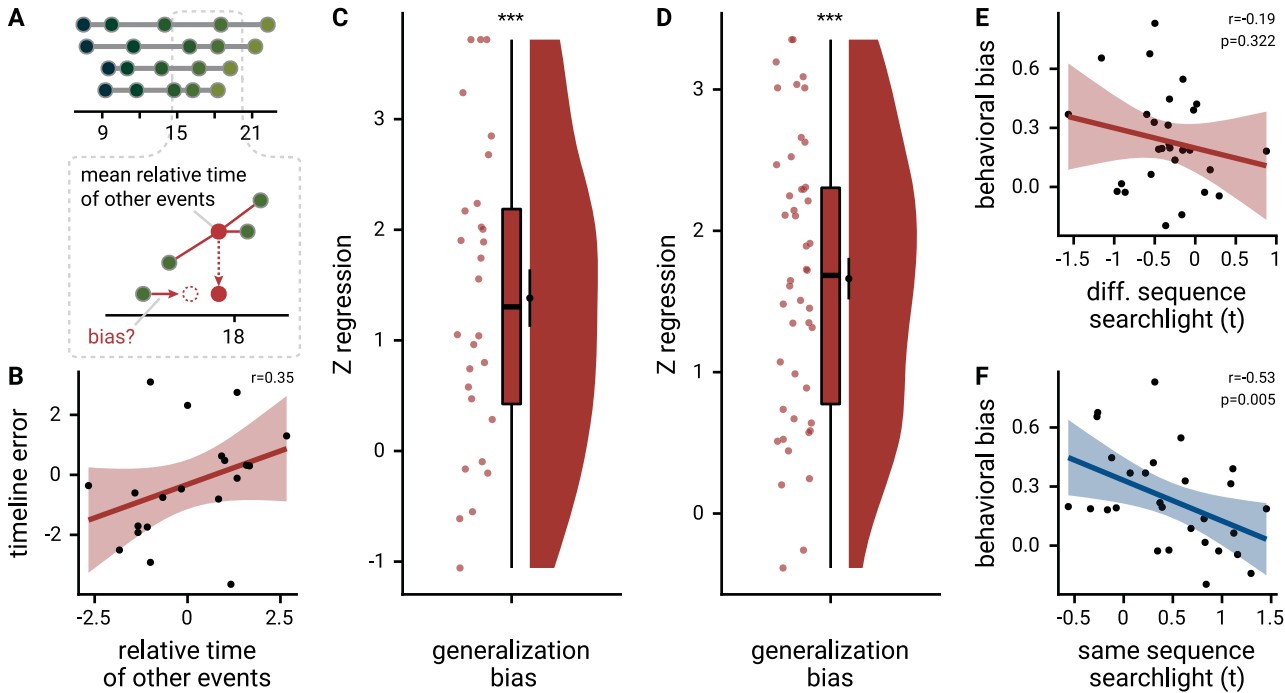

**Fig. 8 Structural knowledge biases construction of event times. A** The generalization bias quantifies the influence of structural knowledge on the construction of individual event times. For each event, the mean time of events at the same sequence position in the other sequences was calculated to test whether event times were biased towards the relative time of other events. **B** The scatterplot illustrates the generalization bias for an example participant. Each circle corresponds to one event and the regression line highlights the relationship between the relative time of other events and the errors in constructed event times. The example participant was chosen to have a median-strength generalization bias. See Supplementary Fig. 10 for the entire sample. Correlation coefficient is based on Pearson correlation. **C** The relative time of events from other sequences predicted signed event time construction errors as measured in the timeline task. Positive values indicate that when other events took place late relative to a specific event, the time of that event was estimated to be later than when other events were relatively early. Circles show individual participant Z-values from participant-specific linear models (**B**); boxplot shows median and upper/lower quartile along with whiskers extending to most extreme data point within 1.5 interquartile ranges above/below the upper/lower quartile; black circle with error bars corresponds to mean ± S.E.M.; distribution shows probability density function of data points. **D** The generalization bias in event time construction through structural knowledge was replicated in an independent sample ($n = 46$) based on Montijn et al.[66]. Data shown as in **C**. **E** The behavioral generalization bias (regression coefficients from summary statistics approach) did not correlate significantly with the across-sequence generalization effect in the anterior hippocampus (searchlight peak voxel t-values). **F** We observed a significant negative correlation between the same-sequence searchlight effect (peak voxel t-values) and the behavioral generalization bias (regression coefficients from summary statistics approach), suggesting that participants with strong hippocampal representations of the temporal relations between events from the same sequence were less biased by structural knowledge in their construction of event times. Statistics in **E** and **F** are based on Spearman correlation. Lines in **E** and **F** show least squares lines, shaded regions correspond to 95% confidence intervals. Source data are provided as a Source Data file. ***$p < 0.001$.

data show that virtual time explained representational changes when competing for variance with order and real time also for events from different sequences. This makes it unlikely that the hippocampal generalization effect was driven exclusively by events at the first or last sequence position. The generalization of temporal distances across sequences in the hippocampus is in line with the contribution of constructive mnemonic processes to flexible cognition via the recombination of elements across experiences and statistical learning[13,40,43,46,48,49,68,69]. More generally, it is consistent with the role of the hippocampus in forming cognitive maps of relational structures and in generalizing structural knowledge to novel situations[12,38,51,53,57,70,71].

Structural knowledge and mnemonic construction are intertwined. In two independent samples, we show that general time patterns, abstracted from other sequences, bias the construction of specific event times. When events at the same sequence position, but in other sequences, took place relatively late to the time of an event, the time of that event was remembered to be later than when the other events occurred relatively early. This generalization bias shows that knowledge about events at structurally similar positions contributes to constructive memory for specific

events. It is in line with biases resulting from the exploitation of environmental statistics when reconstructing stimulus sizes from memory[55,56], when estimating brief time intervals[72,73], or when discriminating the order of previously presented stimuli[74]. Likewise, prior knowledge can distort memories for short narratives[75], spatial associations[76] and temporal positions[62]. Consistent with the suggested role of grid cells in the representation of spatial structure, distortions in mnemonic reconstructions of spatial relations induced through boundary geometry follow predictions from models of grid-cell functioning[77]. Further, recombining information across episodes for associative inference can induce false memories for contextual details[68,78], illustrating that generalization impacts memory for specific associations. In line with the greater reproduction of episodic details by participants whose recall follows the temporal structure of an experience more closely[79], these findings highlight that structural knowledge and mnemonic construction are interwoven. More broadly, abstract semantic or schematic knowledge may provide a scaffold for the recall of episodic details[4,7,38,80,81]. Our findings show that structural knowledge not only facilitates, but also biases constructive memory.

The way temporal relations shaped hippocampal multi-voxel pattern similarity differed between pairs of events from the same and different sequences. We observed positive correlations between temporal distances and hippocampal representational change, which were characterized by relatively decreased pattern similarity for nearby compared to increased pattern similarity for more distant events from the same sequence. One possible explanation for the surprising direction of this effect could be that, compared to our previous work where participants encountered only one sequence[21], participants relied more on associative encoding strategies when learning multiple sequences in the present experiment. Possibly, the need to link events belonging to the same sequence altered how pattern similarity changes relate to temporal distances for these same-sequence events. In line with this interpretation, prior work has shown that the relationship of hippocampal pattern similarity and temporal memory can depend on factors like the use of associative encoding strategies and the presence of event boundaries marking switches between sequences of images from the same category[22,24,82]. Successful recency discrimination was associated with more similar hippocampal representations during encoding when participants were encouraged to use associative strategies to encode the order of image sequences from two alternating visual categories[22]. A different study found more dissimilar hippocampal representations for stimuli whose order was later remembered correctly[24]. Thus, the formation of associations between same-sequence events could explain why correlations of pattern similarity change were, in contrast to our previous work[21], positive. A second possible interpretation of this effect is based on observations that the hippocampus differentiates similar episodes[47,83–86]. Hippocampal differentiation could explain the relative decrease of pattern similarity for temporally close events from the same sequence. However, the generalization across sequences does not directly follow from a differentiation account.

The hippocampus supports constructive memory and generalization in concert with a distributed network of brain regions. In addition to medial temporal lobe structures, the mental simulation of past and future episodic scenarios recruits a core network including medial prefrontal and retrosplenial cortex as well as lateral parietal and temporal areas[39,87]. Notably, this network overlaps with areas supporting the recombination of elements and generalization. For example, both the construction of novel experiences based on the combination of multiple elements[88] and memory integration across episodes[47] are supported by the medial prefrontal cortex and the hippocampus. In sequence processing, representational similarity is increased for items occupying the same position in different sequences in parahippocampal, retrosplenial and medial prefrontal cortices as well as in the angular gyrus[18,89]. Likewise, sequence positions can be decoded from magnetoencephalographic responses elicited by visual stimuli presented in scrambled order[90]. In line with the suggestion that the posterior parietal cortex supports generalization by projecting stimuli onto a low-dimensional manifold[91], neural magnitude representations that generalize across task contexts have been observed using EEG[63,92]. While we did not observe effects outside the hippocampal-entorhinal region that survived corrections for multiple comparisons, we note that, based on our prior hypotheses, we opted for high-resolution coverage of the medial temporal lobe at the cost of reducing the field of view of our MR images. As the events in our task can be conceived of as being arranged along one or multiple, parallel mental number lines, future research could test how the parietal cortex encodes event relations to explore commonalities with and differences to the generalization of event times observed in the hippocampus.

Our paradigm allows a highly controlled read-out of representational change relative to a pre-learning baseline scan. Events are shown in the same random order before and after learning, ruling out that prior associations or the temporal auto-correlation of the blood-oxygen-level-dependent signal drives our effects. Future studies could extend the paradigm to investigate how hierarchically nested sequences are represented, for example by introducing higher-order relations between sequences – akin to different days being grouped in weeks. The precise temporal dynamics of the generalized hippocampal event representation pose another intriguing question. Based on the report that the temporal organization of memory reactivation relative to the hippocampal theta phase reflects semantic relations between items[93], a speculative hypothesis is that a theta phase code could also underlie memory for temporal relations of events from the same and different sequences.

In conclusion, our findings show that the similarity of event representations in the hippocampus reflects relations between events that go back to mnemonically constructed event times, highlighting the impact of mnemonic construction on sequence memory beyond the effects of event order and real elapsing time. Temporal relations are generalized to events from different sequences, in line with hippocampal contributions to the abstraction of structural knowledge and the generalization across episodes. General time patterns abstracted from other sequences systematically influence the construction of specific event times, demonstrating that constructions of specific scenarios build on structural knowledge.

## Methods

**Participants**. 31 participants were recruited for this experiment. Participants gave written informed consent prior to participation. All proceedings were approved by the local ethics committee (CMO Regio Arnhem-Nijmegen). One participant aborted the experiment due to feeling claustrophobic when entering the MR scanner. Two participants were excluded from further analysis due to bad memory performance and technical difficulties during data acquisition. Thus, the sample consisted of 28 participants (21 female, age: mean±standard deviation 23.04 ± 3.21 years, range 18–31 years).

### Procedure

*Overview*. The experiment consisted of four parts (Fig. 1A) and lasted approximately 2.5 h in total. The first three parts were performed inside the MR scanner and comprised a learning task lasting around 50 min that was completed in between two blocks of a picture viewing task of around 25 min each. The tasks inside the scanner were presented on a rear-projection screen with a resolution of 800 × 600 pixels and implemented using Presentation (version 16.2, Neurobehavioral Systems). Subsequently, outside of the scanner, participants performed two short memory tasks in front of a computer screen, implemented with custom Matlab code. The tasks are described in more detail below. Data analysis was carried out using FSL (version 5.0.4)[94] and R (version 3.6.1)[95].

*Stimuli*. The stimuli (Supplementary Fig. 1) used throughout the experiment were created within the life-simulation computer game The Sims 3 (Electronic Arts) by taking screenshots. Each image featured a scene in the life of an affluent family. The main character, the family father, was visible in all scenes. In addition, the mother, son, daughter and family dog appeared in some of the images. All of the depicted events took place within the same family home, but showed activities in a number of different rooms. In an effort to design stimuli with minimal to no indication of day time, the house had constant artificial lighting, but no windows or clocks. The 21 pictures used in this study were selected from an initial set of 35 pictures based on an independent sample rating them as the most ambiguous with regard to the time of day they could take place. One image served as a target image for the picture viewing tasks (see below), while the other 20 event images were randomly assigned to different times and days for every participant.

*Picture viewing tasks*. In the picture viewing tasks (Fig. 1B), participants viewed a stream of the event images. Their task was to look at the images attentively and to respond via button press whenever a target picture, which showed the father feeding the family's dog, was presented (pre-learning: 95.71% ± 7.90% mean ± standard deviation of percentage of hits; 881.34 ms ± 131.43 ms mean ± standard deviation of average reaction times; post-learning: 95.71% ± 6.90% mean ± standard deviation of percentage of hits; 841.40 ms ± 162.16 ms mean ± standard

deviation of average reaction times). The task consisted of 10 mini-blocks. In each mini-block, the target image and the 20 images, which would later make up the virtual days (see Day Learning Task), were shown in random order. Mini-blocks were separated by breaks of 15 s. Stimulus presentations lasted 2.5 s and were time-locked to fMRI volume acquisition onsets. Scene stimuli within a mini-block were separated by 2 or 3 repetition times (TR), randomly assigned so that both stimulus onset asynchronies occurred equally often.

For each participant, we generated a random stimulus order with the constraint that no scene was consistently presented at early or late positions across mini-blocks. Specifically, we compared sequence positions across mini-blocks between the images using a one-way ANOVA. We discarded randomizations where this ANOVA was statistically significant to exclude biases in presentation order. Crucially, the same, participant-specific random order of stimuli and inter-stimulus intervals was used in both the pre-learning and the post-learning picture viewing task. Thus, any systematic differences in the representational similarity of event pairs between the two picture viewing tasks do not go back to differences in the timing of stimulus presentations or the temporal auto-correlation of the BOLD signal. Rather, we interpret such changes to be a consequence of the learning task.

*Day learning task.* In this task, 20 of the 21 scenes, which were shown in the picture viewing tasks, were presented repeatedly. This time, however, they were grouped into multiple sequences introduced to participants as "virtual days". There were four different sequences, each comprising 5 events. Events from the same sequence were always shown in a specific order and with a specific time delay between them. Scenes were on screen for 1.5 s. At the end of each sequence, an image of a moon was shown for 5 s, then the next sequence began. Every sequence was presented 7 times. There were 7 mini-blocks in this task. Within each of these, every sequence was presented once. At the end of a mini-block, a 30 s break followed, then the next block started. The order in which the sequences were presented differed randomly across the 7 mini-blocks.

We instructed participants that the scenes depicted events from the life of a family and that the sequences of event images corresponded to different days in the family's life. Participants were asked to memorize which scenes made up the different sequences (Fig. 1C). We further instructed them to learn when during the respective sequence each event occurred. Specifically, we asked participants to learn event times relative to a virtual clock. This clock was running hidden from participants and event images were shown whenever the hidden clock reached the specific event time (Fig. 1C, Supplementary Fig. 2A, B). The task was devised such that participants had to rely on their experience of passing real time and mnemonic construction to infer the times of events.

Specifically, to give participants an indication of virtual time, the hidden clock was made visible 6 times for every presentation of a sequence: once before the first event, once in between successive events, and once after the last event. Participants received no cues about elapsing real time, but had to use their experience of passing real time between virtual time cues to infer the event times relative to the hidden virtual clock. Importantly, the exposure of the hidden clock occurred at random times for each sequence presentation (Supplementary Fig. 2C, D), with the constraint that it could not be revealed closer than 2 s to a preceding or subsequent event. Thus, participants saw different time cues in each repetition of a sequence. For example, while a specific event always happened at the same virtual time, e.g. 2:07 p.m., the virtual clock could be exposed at any time before the event, e.g. corresponding to 1:32 p.m. in the first repetition of the sequence, and corresponding to 1:17 p.m. in the second repetition. Because true event times were never revealed, participants could not exclusively rely on associative learning to solve the task. Time cues were visible for 1.5 s, but displayed only the time at the start of exposure, i.e. the displayed time did not change within the duration of its presentation.

In short, participants had to combine their experience-based estimates of passing time with the virtual time cues provided by the exposures of the otherwise hidden clock to infer the time at which each event in each sequence took place. Crucially, we varied the speed of the hidden clock between sequences in an effort to partly dissociate real time (in seconds) from virtual time (in virtual hours). Thus, for two sequences more virtual time passed in a comparable amount of real elapsing time (Fig. 1C, Supplementary Fig. 2). Correlations between the linearly increasing time metrics are inevitably high (Pearson correlation of virtual time with order $r = 0.969$ and virtual time with real time $r = 0.975$). Still this manipulation allowed us to determine using multiple regression whether virtual time explained constructed event times when competing for variance with real elapsed time and event order and whether hippocampal pattern similarity changes related to temporal distances in virtual time beyond ordinal distances and real time distances. Regression models including collinear predictor variables do not result in biased parameter estimates[96,97].

*Sorting task.* The day sorting task (Fig. 1D) was performed in front of a computer screen. The 20 event images from the day learning task were presented on the screen in a miniature version. They were arranged in a circle around a central area displaying 4 rectangles. Participants were instructed to drag and drop all events of the same sequence into the same rectangle with a computer mouse. Participants freely chose which rectangle corresponded to which sequence as the sequences were not identifiable by any label and were presented in differing orders across mini-blocks during learning.

*Timeline task.* In this task, participants saw a timeline ranging from 6 a.m. to midnight together with miniature versions of the five event images belonging to one sequence (Fig. 1E). Participants were instructed to drag and drop the event images next to the timeline so that scene positions reflected the event times they had inferred in the day learning task. To facilitate precise alignment to the timeline, event images were shown with an outward pointing triangle on their left side, on which participants were instructed to base their responses.

**MRI acquisition.** MRI data were recorded with a 3 T Siemens Skyra scanner (Siemens, Erlangen, Germany). A high-resolution 2D EPI sequence was used for functional scanning (TR = 2270 ms, TE = 24 ms, 40 slices, distance factor 13%, flip angle 85°, field of view (FOV) 210 x 210 x 68 mm, voxel size 1.5 mm isotropic). The field of view (FOV) was aligned to fully cover the medial temporal lobe, parts of ventral frontal cortex and (if possible) calcarine sulcus. Functional images for the two picture viewing tasks and the learning task were acquired in three runs. In addition to these partial-volume acquisitions, 10 scans of a functional whole-brain sequence were also acquired to improve registration during preprocessing. The sequence settings were identical to the functional sequence above, but instead of 40 slices, 120 slices were acquired, leading to a longer TR (6804.1 ms). A structural scan was acquired for each participant (TR = 2300 ms; TE = 315 ms; flip angle = 8°; in-plane resolution = 256 × 256 mm; number of slices = 224, voxel resolution = 0.8 × 0.8 × 0.8 mm). Lastly, a gradient field map was acquired (for $n = 21$ participants only due to time constraints), with a gradient echo sequence (TR = 1020 ms; TE1 = 10 ms; TE2 = 12.46 ms; flip angle = 90°; volume resolution = 3.5 × 3.5 × 2 mm; FOV = 224 × 224 mm).

**ROI definition.** Our previous work demonstrates representations reflecting the temporal relations of events from one sequence in the anterior hippocampus[21] and the anterior-lateral entorhinal cortex[27]. More generally, these regions have been implicated in temporal coding and memory (for review, see ref. [10]). Further, the hippocampus has been linked to inferential reasoning and generalization[46,48,49,51,53]. We thus focused our analyses on these regions. Region of interest (ROI) masks were based on participant-specific FreeSurfer segmentations (version 6.0.0–2), which yielded masks for the entire hippocampus and entorhinal cortex. These were co-registered to participants' functional space. We defined anterior hippocampus using the Harvard-Oxford atlas mask (thresholded at 50% probability), selecting all voxels anterior to MNI $y = -21$ based on Poppenk et al.[98]. The resulting anterior hippocampus mask was also co-registered to participants' functional space and intersected with the participant-specific hippocampal mask from FreeSurfer. The mask for the anterior-lateral entorhinal cortex was based on Navarro Schröder et al.[99]. It was co-registered to participants' functional space and intersected with the entorhinal cortex mask from FreeSurfer.

**Data analysis**

*Behavioral data analysis*

Sorting task: For analysis of the sorting task, we took the grouping of event images as provided by the participants and assigned them to the four sequences to ensure maximal overlap between actual and sorted sequence memberships. While the assignment of groupings to sequences is unambiguous when performance is, as in our sample, high, this procedure is potentially liberal at lower performance levels. We then calculated the percentage of correctly sorted event images for each participant, see the raincloud plot[100] in Fig. 2A.

In an exploratory analysis, we searched for systematic errors in the sorting task. Specifically, we looked for swap errors where participants interchanged events occurring at the same position between two or more sequences. We used a $\chi^2$-test to assess whether the number of swap errors deviated from uniformity across sequence positions. To test whether participants made more swap errors than expected from chance we ran a permutation test where we introduced sorting errors for randomly selected events. For each of 10 000 iterations, we generated a surrogate sample of sorting results with the number of randomly introduced sorting errors matching the number of errors made by the different participants in our sample. We then quantified the proportion of swap errors across this surrogate sample. This resulted in a distribution of the proportion of swap errors that would be expected from random sorting errors. We assessed how many permutations yielded proportions of swap errors larger or equal to the proportion of swap errors observed in the fMRI sample to compute a p-value and further quantified a z-value as the difference between the observed swap error proportion and the mean of the chance distribution divided by the standard deviation of the chance distribution. We tested whether the number of swap errors was related to absolute errors in the timeline task (see below) using Spearman's correlation and a $t$-test for independent samples.

Timeline task: We analyzed how well participants constructed the event times based on the day learning task. We quantified absolute errors across all events (Fig. 2C) as well as separately for the five sequence positions (Fig. 2D), the four sequences (Supplementary Fig. 3A) and as a function of virtual clock speed (Supplementary Fig. 3B). Using two approaches we tested whether virtual time drove participants' responses rather than the sequence order or objectively elapsing

time. For the summary statistics approach, we ran a multiple regression analysis for each participant with virtual time, sequence position (order), and real time since the first event of a day as predictors of responses in the timeline task. To test whether virtual time indeed explained participants' responses even when competing for variance with order and real time, included in the model as control predictors of no interest, we compared the participant-specific $t$-values of the resulting regression coefficients against null distributions obtained from shuffling the remembered times against the predictors 10,000 times. We converted the resulting $p$-values to $Z$-values and tested these against zero using a permutation-based $t$-test (two-sided; $\alpha = 0.05$; 10,000 random sign-flips, Fig. 2E). As a measure of effect size, we report Cohen's d with Hedges' correction and its 95% confidence interval as computed using the effsize-package[101].

Second, we addressed this question using linear mixed effects modeling. Here, we included the three $z$-scored time metrics as fixed effects. Starting from a maximal random effect structure[102], we simplified the random effects structure to avoid convergence failures and singular fits. The final model included random intercepts and random slopes for virtual time for participants. The model results are visualized by dot plots showing the fixed effect parameters with their 95% confidence intervals (Supplementary Fig. 4A) and marginal effects (Supplementary Fig. 4B) estimated using the ggeffects package[103]. To assess the statistical significance ($\alpha = 0.05$) of virtual time above and beyond the effects of order and real time, we compared this full model to a nested model without the fixed effect of virtual time, but including order and real time, using a likelihood ratio test. Supplementary Table 1 provides an overview of the final model and the model comparison.

To explore whether structural knowledge about general time patterns biases the construction of event times, we assessed errors in remembered event times. Specifically, when constructing the time of one specific event, participants could be biased in their response by the times of the events from other sequences at that sequence position. For each event, we quantified the average time of events in the other sequences at the same sequence position (Fig. 8A). For example, for the fourth event of the first sequence, we calculated the average time of the fourth events of sequences two, three and four. We then asked whether the deviation between the average time of other events and an event's true virtual time was systematically related to signed errors in constructed event times. A positive relationship between the relative time of other events and time construction errors indicates that, when other events at the same sequence position are relatively late, participants are biased to construct a later time for a given event than when the other events took place relatively early. In the summary statistics approach, we ran a linear regression for each participant (Fig. 8B, Supplementary Fig. 10A) and tested the resulting coefficients for statistical significance using the permutation-based procedures described above (Fig. 8C). The regression coefficients from this approach were used to test for a relationship between the behavioral generalization bias and the hippocampal searchlight effects (see below). Further, we analyzed these data using the linear mixed model approach (Supplementary Fig. 4O, P, Supplementary Table 16).

To replicate the results from this exploratory analysis, we conducted the same analysis in an independent group of participants. These participants ($n = 46$) constituted the control groups of a behavioral experiment testing the effect of stress induction on temporal memory[66]. They underwent the same learning task as described above with the only difference being the duration of this learning phase (4 rather than 7 mini-blocks of training). The timeline task was administered on the day after learning. The procedures are described in detail in Montijn et al.[66]. The data from this independent sample are shown in Fig. 8D and Supplementary Fig. 10B.

*MRI preprocessing.* Preprocessing was performed using FSL FEAT (version 6.00). Functional scans from the picture viewing tasks and the whole-brain functional scan were submitted to motion correction and high-pass filtering using FSL FEAT. For the two picture viewing tasks, data from each mini-block was preprocessed independently. For those participants with a field map scan, distortion correction was applied to the functional data sets. No spatial smoothing was performed. Functional images from the two picture viewing tasks were then registered to the preprocessed mean image of the whole-brain functional scan. The whole-brain functional images were registered to the individual structural scans. The structural scans were in turn normalized to the MNI template (1 mm resolution). Gray matter segmentation was done on the structural images, and the results were mapped back to the space of the whole-brain functional scan for later use in the analysis.

*Representational similarity analysis.* Representational similarity analysis (RSA)[104] was first implemented separately for the pre-and post-learning picture viewing task. It was carried out in ROIs co-registered to the whole-brain functional image and in searchlight analyses (see below). For the ROI analyses, preprocessed data were intersected with the participant-specific anterior hippocampus and ante-rolateral entorhinal cortex ROI masks as well as a brain mask obtained during preprocessing (only voxels within the brain mask in all mini-blocks were analyzed) and the gray matter mask. For each voxel within the ROI mask, motion parameters from FSL MCFLIRT were used as predictors in a general linear model (GLM) with the voxel time series as the dependent variable. The residuals of this GLM (i.e. data that could not be explained by motion) were taken to the next analysis step. As the presentation of images in the picture viewing tasks was locked to the onset of a new

volume (see above), the second volume after image onset was selected for every trial, effectively covering the time between 2270 and 4540 ms after stimulus onset. Only data for the 20 event images that were shown in the learning task were analyzed; data for the target stimulus were discarded. The similarity between the multi-voxel activity pattern for every event image in every mini-block with the pattern of every other event in every other mini-block was quantified using Pearson correlation coefficients. Thus, comparisons of scenes from the same mini-block were excluded. Next, we calculated mean, Fisher z-transformed correlation coefficients for every pair of events, yielding separate matrices of pattern similarity estimates for the pre- and the post-learning picture viewing tasks (Fig. 3).

In order to assess changes in representational similarity between the two picture viewing tasks, we quantified pattern similarity changes as the difference of the respective correlation coefficients for every pair of events between the post-learning picture viewing task and its pre-learning baseline equivalent (Fig. 3). Then, we analyzed how these difference values related to temporal relations between events, which we quantified using the absolute distances in virtual time ("virtual time") between events (Fig. 1C, bottom right). We further tested whether the effect of virtual time on anterior hippocampal pattern similarity change persisted when including the absolute difference between sequence positions ("order") and the interval in seconds between events ("real-time") as control predictors of no interest in the model. Time metrics were z-scored within each participant prior to analysis. We separately tested the effect of virtual time for event pairs from the same or different sequences and used a Bonferroni-corrected α-level of 0.025 for these tests. To implement these tests, we employed two approaches to model-based RSA that are described in detail below. We used a summary statistics approach, which uses permutation-based procedures on the subject level as well as on the group level, in line with recommendations for the analysis of multi-voxel patterns[105]. We also implemented our statistical analyses using linear mixed effects models, which capture within-subject dependencies using random effects while estimating the fixed of interest on all data points. Mixed effects models are well-suited to test more complex interactions. The fact that the results of the two analysis approaches converge demonstrates that our findings are robust to the specific statistical technique. We used an α-level of 0.05 for both approaches because they are not independent as they are implemented on the same data and test the same hypotheses.

Summary statistics approach: In the summary statistics approach, we used the different time metrics as predictors for pattern similarity change. We set up a GLM with the given variable from the day learning task as a predictor and the pairwise representational change values as the criterion for every participant. The $t$-values of the resulting model coefficients were then compared to a null distribution obtained from shuffling the dependent variable of the linear model (i.e. pattern similarity change) 10,000 times. This approach to permutation-testing of regression coefficients controls Type I errors even under situations of collinear regressors[106]. Resulting $p$-values for each coefficient were transformed to a Z-score. The Z-scores were then used for group-level inferential statistics.

Group-level statistics were carried out using permutation-based procedures. For $t$-tests, we compared the observed $t$-values against a surrogate distribution obtained from 10,000 random sign-flips to non-parametrically test against 0 or to assess within-participant differences between conditions (two-sided tests; $\alpha = 0.05$ unless stated otherwise). We report Cohen's d with Hedges' correction and its 95% confidence interval as computed using the effsize-package for R. For paired tests, Cohen's d was calculated using pooled standard deviations and confidence intervals are based on the non-central $t$-distribution. Permutation-based repeated measures ANOVAs were carried out using the permuco-package[107] and we report generalized $\eta^2$ as effect sizes computed using the afex-package[108].

Linear mixed effects: Second, we employed linear mixed models to assess how learned sequence relationships were reflected in pattern similarity change using the lme4 package[109]. Mixed models have the advantage of estimating fixed effects and their interactions using all data, rather than performing inferential statistics on just one value per participant. We used the different time metrics as the fixed effects of interest. Factorial predictors (region of interest: anterior hippocampus and anterior-lateral entorhinal cortex; sequence: same vs. different) were deviation-coded. Within-subject dependencies were captured using random effects. Following the recommendation by Barr et al[102]., we always first attempted to fit a model with a maximal random effects structure including random intercepts and random slopes for participants. If these models did not converge or resulted in singular fits, we reduced the random effects structure. We always kept random slopes for the fixed effect of interest in the model to avoid anti-conservativity when testing fixed effects or their interactions[102,110]. The mixed effects models were fitted using maximum likelihood estimation.

We assessed the statistical significance of fixed effects of interest using likelihood ratio tests ($\alpha = 0.05$). Specifically, the model including the fixed effect of interest was compared against a nested, reduced model excluding this effect, but with the same random effects structure. Throughout the manuscript we report the results of these model comparisons ($\chi^2$-tests with one degree of freedom) and refer to supplemental tables for summaries of the final mixed model parameters. We visualize fixed effect estimates with their 95% confidence intervals as dot plots and further illustrate effects using estimated marginal means[103].

Multidimensional scaling: We aimed to explore how hippocampal event representations of the different sequences could be embedded in a low-dimensional representational space to give rise to the positive and negative correlations of pattern similarity change and temporal distances for same-sequence and different-sequence events, respectively. For each pair of events, we generated an expected similarity value (Supplementary Fig. 6D) using the fixed effects of the mixed model fitted to hippocampal pattern similarity that captures the interaction between virtual temporal distances and sequence membership (c.f. Fig. 5, Supplementary Fig. 4I, J, and Supplementary Table 7). Using the predict-method implemented in the lme4-package[109], we generated model-derived similarity values for all event pairs given their temporal distances and sequence membership. We chose this approach over the raw pattern similarity values to obtain less noisy estimates of the pairwise distances. Using the smacof-package[111], the model-predicted similarities were converted to distances and the resulting distance matrix (Supplementary Fig. 6D) was subjected to non-metric multidimensional scaling using two dimensions. We chose two dimensions to be able to intuitively visualize the results. Because MDS is sensitive to starting values, we ran multidimensional scaling 1000 times with random initial configurations and visualized the resulting configuration with the lowest stress value. Basing this analysis on the model-derived similarities assumes the same relationship of virtual temporal distances for all event pairs from different sequences, but we would like to note that not all solutions we observed, in particular those with higher stress values, resulted in parallel configurations for the four sequences.

We tested the stress value of the resulting configuration against a surrogate distribution of stress values obtained from permuting the input distances on each of 1000 iterations. Using the mean and standard deviation of the resulting null distribution, we obtained a z-value as a test statistic and report the proportion of stress values in the null distribution that were equal to or smaller than the observed stress value (Supplementary Fig. 6E). Additionally, we contrasted the distances between pairs of events in the resulting configuration between distances separated by high or low (median split) input distances using a t-test for independent samples (Supplementary Fig. 6F). Using a Spearman correlation, we quantified the relationship of the input distances and the distances in the resulting configuration (Supplementary Fig. 6G).

Searchlight analysis: We further probed how temporal distances between events shaped representational change using searchlight analyses. Using the procedures described above, we calculated pattern similarity change values for search spheres with a radius of 3 voxels around the center voxel. Search spheres were centered on all brain voxels within our field of view. Within a given search sphere, only gray matter voxels were analyzed. Search spheres not containing more than 25 gray matter voxels were discarded. For each search sphere, we implemented linear models to quantify the relationship between representational change and the learned temporal structure. Specifically, we assessed the relationship of pattern similarity change and absolute virtual temporal distances, separately for event pairs from the same sequences and from pairs from different sequences. In a third model, we included all event pairs and tested for an interaction effect of sequence membership (same or different) predictor and virtual temporal distances. The t-values of the respective regressors of interest were stored at the center voxel of a given search sphere.

The resulting t-maps were registered to MNI space for group level statistics and spatially smoothed (FWHM 3 mm). Group level statistics were carried out using random sign flipping implemented with FSL Randomise and threshold-free cluster enhancement. We corrected for multiple comparisons using a small volume correction mask including our a priori regions of interest, the anterior hippocampus and the anterior-lateral entorhinal cortex. Further, we used a liberal threshold of $p_{uncorrected} < 0.001$ to explore the data for additional effects within our field of view. Exploratory searchlight results are shown in Supplementary Fig. 9 and clusters with a minimum extent of 30 voxels are listed in Supplementary Tables 12, 14 and 15.

To test whether within- and across-sequence representations overlap, we defined an ROI based on the within-sequence searchlight analysis. Specifically, voxels belonging to the cluster around the peak voxel, thresholded at $p < 0.01$ uncorrected within our small volume correction mask, were included. The analysis of representational change was then carried out as described for the other ROIs above. The results observed using a threshold of $p < 0.001$ were not statistically different from those obtained with a threshold of $p < 0.01$ ($t_{27} = -0.95$, $p = 0.338$; test against 0 using the ROI resulting from the $p < 0.001$ threshold: $t_{27} = -1.98$, $p = 0.056$).

Relationship to behavior: We used the regression coefficients quantifying the strength of the behavioral generalization bias to test for an across-subject relationship with the RSA searchlight effects. For each participant, we extracted the t-value of the across-sequence and the within-sequence searchlight effects from the peak voxel in our a priori regions of interest. We chose this approach because the searchlight analyses provide greater spatial precision than anatomically defined region of interest masks. We used Spearman correlations to test for a relationship of the RSA searchlight effects and the behavioral generalization bias ($\alpha = 0.025$, corrected for two comparisons).

**Reporting summary**. Further information on research design is available in the Nature Research Reporting Summary linked to this article.

## Data availability

Data to reproduce the statistical analyses reported in this paper are available on the Open Science Framework[112] (https://osf.io/zxnc8/). Source data are provided with this paper.

## Code availability

Analysis code and documentation are available on GitHub (https://jacbel.github.io/virtem_code/)[113].

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

## Acknowledgements

The authors would like to thank Ignacio Polti for helpful discussions on the behavioral generalization bias, Iris M. Engelhard for making the data of the replication sample available, and Roland Benoit for helpful comments on a previous version of the manuscript. This work was supported by the Netherlands Organisation for Scientific Research (NWO-Vidi 452-12-009; NWO-MaGW 406-14-114), the European Research Council (ERC-CoG GEOCOG 724836) and the Max Planck Society. C.F.D.'s research is further supported by the Kavli Foundation, the Centre of Excellence scheme of the Research Council of Norway—Centre for Neural Computation (223262/F50), The Egil and Pauline Braathen and Fred Kavli Centre for Cortical Microcircuits, the Jebsen Centre for Alzheimer's Disease, and the National Infrastructure scheme of the Research Council of Norway—NORBRAIN (197467/F50). N.D.M. was funded by the Netherlands Organization for Scientific Research (NWO 453-15-005, awarded to Iris M. Engelhard) during the acquisition of the data used for the replication of the behavioral generalization effect.

## Author contributions

L.D. and C.F.D. conceived the experiment. L.D. and N.D.M. developed the tasks and acquired the data. L.D. preprocessed the MRI data. J.L.S.B., L.D. and C.F.D. conceived the analyses. J.L.S.B. analyzed the data and wrote the manuscript with input from C.F.D. All authors discussed the results and contributed to the paper.

## Funding

## Competing interests

The authors declare no competing interests.
