## [Peer Review File · Nature Communications]

Mnemonic construction and representation of temporal structure in the hippocampal formationREVIEWER COMMENTS

Reviewer #1 (Remarks to the Author):

The paper by Bellmund and colleagues describes a human fMRI study in which subjects learned various different sequences of events (images taken from a virtual environment). The event sequences were described as different “days” in the virtual world and even sequence had its own virtual time. Critically, virtual time covaried with actual time, but was partly dissociable because each day had a different “speed.” fMRI data were used to compare the neural representations of the individual events before and after learning and a behavioral post-test was used to measure the accuracy with which subjects remembered the different sequences and, most critically, memory for the timing of events relative to the virtual clock. There are several interesting findings in the paper. Behaviorally, subjects successfully learned virtual times, which is notable because the virtual time for events had to be inferred by integrating the actual elapsed time with the virtual clock (which was never shown when the events actually occurred). Additionally, behavioral constructions of virtual time exhibited some biases from the normative temporal structure. In other words, memory for a given sequences timing was biased by the timing of other events. fMRI data comparing representational structure pre and post learning revealed that the hippocampus showed two forms of learning related change. (1) representations of events from the same sequence changed such that events nearby in time became relatively less similar and (2) representations of events from different sequences but at similar timepoints (within the sequence) become relatively more similar. This difference was not see in the entorhinal cortex, where there was a representation of virtual time that did not differ for within vs. across sequence comparisons. Thus, the hippocampus showed a separation of events within a sequence, while simultaneously exhibited a representation of temporal structure that generalized across different sequences. Hippocampal representations of virtual time (within a sequence) were also related to behavioral biases in that greater hippocampal sequence effects predicted less bias from other sequences.

Overall, this is a very exciting manuscript. The experimental design and analyses are very clever and there are multiple interesting results. I have a few comments and questions, but my sense is that these are all addressable points and that the manuscript is likely to be of broad interest to readers of this journal.

Comments

1. On my first pass through the manuscript, it was hard for me to understand the direction of the effects reported in Figure 4. I assume that a positive effect meant that events nearby in time became more similar. There is a statement on page 7 that says: “Thus, pattern similarity was higher for event pairs separated by longer temporal distances than for pairs separated by shorter intervals.” I actually thought this must have been a type-o (and that it was supposed to say that pattern similarity was higher for SHORTER temporal distances). It was only when I got to figure 5 that I fully realized that the within sequence effects are quite surprising (opposite to what one would likely predict). To prevent similar confusion for readers, I would recommend making some edits. The visualization in 5D is helpful for visualizing the direction of the relationship. So, one option would be to show something similar in figure 4. Or, even just to emphasize (in the text) that the result was surprising. But, from the Introduction of the paper,

there is nothing that motivates this idea or leads the reader to expect it. I am not suggesting that the Introduction be modified to “predict” this result, but I am just flagging that I think readers will be confused.

2. Related to point 1: It’s a little confusing that “negative correlations between representational change and temporal distances” are actually in the prediction you would expect. Negative makes one think it’s going to be the opposite of what would be expected.

3. It was not entirely clear whether it was possible to infer any of the sequence or event structure from the images prior to the day learning task. E.g., do events from the same sequence have any greater visual or conceptual similarity (in terms of rooms, characters, objects, etc.)? This is not necessarily a problem given the pre vs. post design, but it would be useful to know.

4. Would a similar interaction between temporal distance and sequence (within vs. across) be found if sequence position or actual clock time were used instead of virtual time? I don’t know that it is a problem if the answer is “yes,” but it would be interesting to know. Clearly subjects did learn the virtual times, but it is not clear that the surprising within-sequence temporal effects critically depend on virtual time.

5. Two potential accounts of the reduction in similarity for nearby, same-sequence events are offered. One is “These findings might support the view that the formation of associations between elements of a sequence may have altered how pattern similarity changes relate to temporal distances for same-sequence events.” I had difficulty understanding this argument. It would be helpful to flesh this out some more. The other account is a differentiation account. The current results do have a striking similarity to other examples of hippocampal differentiation of similar experiences. However, it is stated that “this interpretation does not directly account for the overall correlation of temporal distances and representational change.” I think this statement is true, but again could be fleshed out slightly more. I imagine the point is that a differentiation account does not explain the changes for across sequence representations?

6. One potential account of the generalization results is that they may reflect some form of primacy or recency effects within the sequences. Namely, events that were presented at the beginning or end of the sequence might be better remembered. Or the first and last events might be more likely to trigger associative memory for the whole sequence—i.e., these events might be better memory cues for the sequence than events in the middle of the sequence. In either of these cases, the hippocampal response to 1st or last images would likely be different in a way that generalizes across sequences but without actually reflecting something about abstract time or position. In other words, you could observe the generalization effects even if the true nature of the hippocampal representation has nothing to do with abstract sequence position—it is just something that is correlated with sequence position. I do not see any obvious way to rule out this account. While I do view this as a limitation, I don’t think this is a fatal flaw. Rather, I think this limitation should be acknowledged in the Discussion.

7. The within sequence hippocampal effects are reminiscent of a recent preprint from Sherman, DuBrow, Winawer, and Davachi <https://www.biorxiv.org/content/10.1101/2021.08.03.454949v1.full.pdf> which found that

hippocampal pattern similarity within an event sequence was actually lower than across sequences. It might be worth acknowledging/citing this paper to add some conceptual support for the idea that the hippocampus differentiates events within a sequence.

8. Related to the MDS reconstruction analysis:

(a) The reconstructions are really striking. But is this based on the change from pre to post learning or just based on the post learning results?

(b) Also, I initially assumed this was based on the raw pattern similarity values and was stunned by the consistency of the reconstructions across different sequences. But it seems the MDS was actually applied to modeled data. I did not entirely follow how this was done and, therefore, it is hard to ascertain whether the remarkable consistency of the reconstructions across sequences simply reflects that structure imposed by the model?

(c) It's a bit hard to reconcile the idea that the hippocampus 'learns' the sequence representation while also systematically showing decreased similarity for nearby events compared to far away events. These ideas seem to be at odds. Seeing the the c-shape structure in the reconstructions really helps provide some intuition for this in that it seems that the start point and end point end up being close(r) together in representational space. Though, it is hard to tell (by eye) whether this reconstruction actually captures this effect. If you just compute the Euclidean distance between the reconstructed pairs, are temporally nearby events actually farther away (greater Euclidean distance) than temporally distant events?

Minor:

9. Line 270: "When events at the same sequence position, but in different sequences, took place late relative to an event...". Is this supposed to say "relative to a SEQUENCE"? or maybe "relative to other events at the same sequence position"?

10. Page 8: "we next assessed whether this effect was driven by the constructed event times beyond sequence order and real time. We thus included the two additional time metrics as control predictors in the model." I initially interpreted this to mean that the behaviorally-constructed times from the post-test were used. But I think that what the authors might have meant is just the "virtual time." If so, I would recommend strictly using "virtual time" to refer to the objective virtual clock given the potential to confuse the objective virtual time with the behaviorally-constructed virtual time. I do understand that virtual time HAS to be constructed, but, still, there is the potential for confusion.

Reviewer #2 (Remarks to the Author):

Bellmund and colleagues report a functional MRI study in which they sought to shed light on the type of temporal memory (i.e. inferred/constructed time, elapsed real time, temporal order) that anterior hippocampal and anterior lateral entorhinal representations encode in support of event sequence memory. Using a representational similarity approach together with a behavioural task that required participants to learn the temporal structure of four different event sequences that unfolded in relation to a 'virtual clock', it was found that anterior hippocampal representations contained information about constructed time for both within and between sequence events. Specifically, events that were further apart in virtual time within the same sequence were associated with greater pattern similarity compared to events that were closer together. In contrast, events that occurred at comparable virtual times between sequences were represented more similarly than those that took place at different virtual times. This latter finding points towards the generalization of temporal relations across sequences and reflective of this, participants demonstrated a 'generalization bias' in their task responses, with the estimated virtual time of any given event being influenced by the time of occurrence of other events in other sequences in the same sequence position. In comparison to the anterior hippocampus, representations in the anterior lateral entorhinal cortex did not differentiate between same-sequence and across-sequence temporal relations, with events occurring close together in time being associated with greater pattern similarity.

There has been much interest in how the brain supports the temporal dimension of event sequence memory and as such, the current study is very timely, with the potential to be conceptually important. The behavioural paradigm is clever, providing a means to disentangle constructed time from elapsed time and event order, and broadly speaking, the experimental methods and data analyses are solid. Indeed, the authors are to be commended for their use of multiple analytical approaches (to provide converging findings) and the implementation of certain control analyses to rule out alternative explanations. Overall, I'm quite enthusiastic about this paper but have a number of issues that I would appreciate further clarity on.

1) It is somewhat surprising that a handful of participants performed relatively poorly on the sorting task (in the 40 - 60 % correct range) and, in fact, there was quite a range in performance across participants. Were the poorest performers on the sorting task also those who produced the greatest absolute error on the timeline task? Related to this, it would be interesting to know whether there were any discernible patterns in the errors that participants made on the sorting task. Specifically, swap errors, in which participants mix-up events occupying the same sequence position across sequences (i.e. A1-B2-C1-D1-E1 & A2-B1-C2-D2-E2), may be a product of generalization across sequences and could conceivably be related to the generalization bias (e.g. greater generalization of temporal structure across sequences may lead to a greater number of swap errors and a greater generalization bias).

2) Was there a significant difference in absolute error on the timeline task between the different sequences? While eye-balling Figure 2B suggests that there wasn't, it may still be useful to demonstrate this statistically and show that clock speed had no impact on participants' ability to construct time.

3) When examining the representational similarity between events from different sequences in

the anterior hippocampus (Supplemental Table 5), was an analysis conducted with real time and order included as additional control factors? The same applies for the examination of representational change in the anterior lateral entorhinal cortex (Supplemental Table 7). The findings of these analyses would be important to report.

4) Given that generalization bias can be quantified for each trial, I'm curious as to whether the authors attempted an analysis whereby they examined trial-by-trial fluctuations in BOLD signal in relation to this measure (e.g. within a GLM with generalization bias as a regressor)? The observation of significant anterior HPC involvement in such an analysis would add further weight to the authors' conclusions.

5) It would be useful if the authors could clarify why they used NMDS to visualize the relationship between the different events, sequences and virtual time, as opposed to MDS, and how they explored/determined the optimal number of dimensions to account for their data. Moreover, to further quantify the output of the NMDS analysis, one could consider using k-means clustering to examine the clustering of the different events across the different sequences.

Minor:

In Figures 5D and 6E, it would be useful if the legend could state how temporal distances were classified as 'high' vs. 'low'

It would be helpful for r and p values to be reported in the correlation figures (e.g. Figure 8B, Sup. Figure8).

To test whether within- and across-sequence representations overlap, ROIs were defined using a $p < 0.01$ uncorrected threshold. How was this threshold chosen and do the findings change dramatically if a threshold of $p < 0.001$ uncorrected is adopted?

Reviewer #3 (Remarks to the Author):

In this manuscript, Bellmund and colleagues interrogate the creation of temporal sequences in the anterior hippocampus and entorhinal cortex. They pursue this through the creation of a well designed experimental paradigm wherein subjects watch distinct event sequences from different imaginary days in the life of a family. The authors attempt to dissociate 'virtual time' in the context of the viewed events and the absolute passage of time, doing so by providing participants with intermittent clock time cues indicating virtual time. The speed of these virtual clocks was manipulated such that the passage of virtual time and absolute time varied across simulated days. A pre/post still-frame viewing design is used to ascertain the differences in representational similarity for images that were presented both within the same sequence and across different sequences. Participants structured their estimates of the temporal structure of events strongly in accordance with virtual time, as opposed to mere temporal order or the real passage of time. This was mirrored by RSA results in the anterior hippocampus and anterior-

lateral entorhinal cortex. Furthermore, the authors report that more temporally distant events within the same sequence become similar in the anterior hippocampus, suggesting that this may be due to beginning and end events being more strongly related. Temporally similar events across sequences, on the other hand, become more similar suggesting a realignment of episodes along a common axis. Conversely, the authors report that the entorhinal cortex featured only generalization of sequence structure rather than sensitivity to within versus across sequence information, such as they observed in the hippocampus. Finally, multidimensional scaling approaches were used to visualize low-dimensional embedding of the sequences of events, which I found to be a nice addition over the prior version of the manuscript.

My overall impression is that this is a very well-designed experiment and a fairly novel approach to studying temporal coding in the human medial temporal lobe. The notion of dissociating virtual time from real time is especially interesting (though, as I note below, I have some questions about the extent to which this is compellingly done). However, there are issues with conceptual framing of the results, and clarity about the way this work fits into and adds to the various corners of the literature that the authors cite. These points are listed below.

Lastly, I will note that I reviewed a prior version of this manuscript for a different journal. Though the authors addressed a few of the concerns I previously voiced, the authors did not really contend with the majority of the points raised from my prior review. Thus, I will largely reiterate those points, and note a few places where progress was made in my view over the previous version.

Major Points:

1. In my prior review, I noted that the framing of the paper was a bit opaque in terms of the study's contributions and scope with regards to the broader literature. Unfortunately, the authors do not seem to have changed very much at all from the previous version to address this concern. I was previously and still am struggling a bit with understanding the major contributions this study makes to changing or solidifying our understanding of the temporal organization of memory. A basic take-away from the paper is that the hippocampus and entorhinal cortex encode temporal sequences, which has been well-established (e.g., work from the authors' own group, as well as work by the Eichenbaum, Davachi, Ranganath, Howard, Fortin, and Kahana labs, among many others). Beyond basic sequence coding, the authors show that this temporal coding appears to align with virtual time rather than absolute time. This accords with recent work from Shimbo et al (Science Advances, 2021), and is novel in terms of human behavior and neural signals in the human brain. However, the framing of the manuscript is a bit scattered and does not effectively communicate this. I previously noted that the Discussion was particularly problematic in this regard. In this newer version, several minor changes seem to have been made which have made the Discussion read better. However, I still believe that the basic issue of too many tenuous links to what appear to be 'hot' research topics is detracting from rather than improving this manuscript.

2. Following from the above, at a more nuanced level, the authors show that the hippocampus shows higher pattern similarity for same-sequence items which are farther apart than items that are closer together in time. The opposite pattern is found for different-sequence items. The authors note the following: "In contrast to our previous work, participants studied multiple

sequences. They might have formed strong associations of same-sequence events on top of inferring each event's virtual time, potentially altering how temporal distances affected hippocampal pattern similarity." This is seemingly rephrased from the previous submission, but I am having difficulty understanding this point and do not find it to be an improvement. Given that the opposing pattern of results for same versus different sequences in the hippocampus is a key feature in the data, I think that a clearer discussion of this effect's directionality (especially given that it is perhaps counterintuitive) is necessary.

3. From my prior review, not addressed: An important aspect of this design and of the results is that behavioral and neural data indicate that participants are encoding information on the basis of virtual time, rather than real time. Given the design of the experiment and stimuli, I am unsure if this should be at all surprising. Participants received any external cue whatsoever about virtual time, compared to none about real time. In fact, aside from aiding in one's understanding of how much time has elapsed between virtual clock cues, it does not seem that real time is at all relevant to participants' ability to engage in the task. While I have no doubt that participants are tracking the virtual timing of these events, or that they are doing so more strongly than they are absolute time, I am not sure how meaningful this comparison really is in the context of the experiment. While the authors did vary virtual clock speed in an attempt at dissociating real from virtual time, one can still reasonably argue that real time is uncued and simply not relevant to completing the task, rendering real time relatively uninformative, and a virtual versus real time comparison a strawman. Moreover, one could reasonably argue that participants' understanding of virtual time involves a combination of virtual + real time, further obscuring meaningful comparisons with real time only. I think the manuscript needs to address this issue.

4. From my prior review, not addressed: It is not clear why the authors chose to conduct two classes of analyses (summary statistics and mixed effects models) for every family of data. This is at best somewhat redundant, and at worst raises questions about correction for multiple comparisons when applying these analyses on the same family of data. Some reported effects fall between p-values of 0.025 and 0.05, meaning a simple Bonferroni correction would prove problematic. If both classes of tests are necessary, a clear case should be made for this approach. Otherwise, it may be prudent to choose one approach. If the authors believe that both classes of tests are necessary, and that multiple comparisons correction is unnecessary, this should be convincingly argued.

Minor Points:

5. Following from my final major point above, the figures are somewhat overwhelming, and carry seemingly redundant information. For example, in Figure 4 the authors present plots associated with a univariate regression and then follow with plots associated with a multiple regression. This depends on the authors' preferred solution to the above concern, but I will note that this figure and others would be simpler to parse if they only showed the multiple regression results, as they converge on the same conclusion as univariate regression, and as they are the stronger analysis via simultaneously evaluating multiple factors. I will note (in somewhat ironic contrast to the rest of this comment) that the authors have now added a multidimensional scaling analysis plot over the prior version of the manuscript, which while not adding anything terribly unique to the results, does provide a nice visualization of sequence representation.

6. Throughout the text, the authors refer to sequences in temporal memory being “actively constructed.” Can claims about temporal relations being actively constructed be made based on pre vs. post task RSA? There is clearly some record of task structure in memory, but it is not clear that this is indicative of some active (rather than passive) process, or that the word “active” is really carrying any meaning here. This is a minor and somewhat nitpicky point, but given the repeated use of this phrase, it warrants some clarification or unpacking.

7. The authors a-priori justification of their ROIs could stand to be fleshed out more. While I understand that prior work from this group has highlighted the anterior hippocampus and anterior-lateral entorhinal cortex, the logic for this selection warrants better justification. Especially given that many of the phenomena the authors allude to in the introduction and discussion are often associated with other regions.

8. For Figure 8, why was the searchlight ROI used and not the anatomical ROI?

Reviewer #1

The paper by Bellmund and colleagues describes a human fMRI study in which subjects learned various different sequences of events (images taken from a virtual environment). The event sequences were described as different “days” in the virtual world and even sequence had its own virtual time. Critically, virtual time covaried with actual time, but was partly dissociable because each day had a different “speed.” fMRI data were used to compare the neural representations of the individual events before and after learning and a behavioral post-test was used to measure the accuracy with which subjects remembered the different sequences and, most critically, memory for the timing of events relative to the virtual clock. There are several interesting findings in the paper. Behaviorally, subjects successfully learned virtual times, which is notable because the virtual time for events had to be inferred by integrating the actual elapsed time with the virtual clock (which was never shown when the events actually occurred). Additionally, behavioral constructions of virtual time exhibited some biases from the normative temporal structure. In other words, memory for a given sequences timing was biased by the timing of other events. fMRI data comparing representational structure pre and post learning revealed that the hippocampus showed two forms of learning related change. (1) representations of events from the same sequence changed such that events nearby in time became relatively less similar and (2) representations of events from different sequences but at similar timepoints (within the sequence) become relatively more similar. This difference was not seen in the entorhinal cortex, where there was a representation of virtual time that did not differ for within vs. across sequence comparisons. Thus, the hippocampus showed a separation of events within a sequence, while simultaneously exhibited a representation of temporal structure that generalized across different sequences. Hippocampal representations of virtual time (within a sequence) were also related to behavioral biases in that greater hippocampal sequence effects predicted less bias from other sequences.

Overall, this is a very exciting manuscript. The experimental design and analyses are very clever and there are multiple interesting results. I have a few comments and questions, but my sense is that these are all addressable points and that the manuscript is likely to be of broad interest to readers of this journal.

We would like to thank the Reviewer for their positive evaluation of our manuscript and the constructive comments for further improvements. By addressing these comments, we believe that we have made our manuscript more accessible for the reader. In particular, we have revised our first main fMRI result figure (Figure 4) to include the visualization of pattern similarity changes contrasted between temporally close and far events. Further, we more clearly point out the surprising nature of this within-sequence effect. Additionally, the results of new analyses provide supporting evidence for the interpretation that the differential representation of temporal relations in the hippocampus reflects the constructed virtual temporal relations beyond sequence order and elapsing real time. Overall, we believe that we have further strengthened the manuscript through these changes and additional analyses. Please find our detailed responses below.

Major Comments

Comment 1

On my first pass through the manuscript, it was hard for me to understand the direction of the effects reported in Figure 4. I assume that a positive effect meant that events nearby in time became more similar. There is a statement on page 7 that says: “Thus, pattern similarity was higher for event pairs separated by longer temporal distances than for pairs separated by shorter intervals.” I actually thought this must have

been a type-o (and that it was supposed to say that pattern similarity was higher for SHORTER temporal distances). It was only when I got to figure 5 that I fully realized that the within sequence effects are quite surprising (opposite to what one would likely predict). To prevent similar confusion for readers, I would recommend making some edits. The visualization in 5D is helpful for visualizing the direction of the relationship. So, one option would be to show something similar in figure 4. Or, even just to emphasize (in the text) that the result was surprising. But, from the Introduction of the paper, there is nothing that motivates this idea or leads the reader to expect it. I am not suggesting that the Introduction be modified to “predict” this result, but I am just flagging that I think readers will be confused.

The reviewer comments on the presentation of the same-sequence pattern similarity effect, which was characterized by positive correlations of temporal distances and hippocampal pattern similarity. We agree that the direction of this effect is surprising. We have followed the recommendations of the reviewer to revise our manuscript to provide better guidance for the reader in order to avoid confusion about the direction of this effect. Specifically, we now illustrate the direction of the effect by showing the raw pattern similarity differences between events separated by low vs. high temporal distances in the revised Figure 4 already (new panel C). Further, we explicitly state that the direction of this effect was surprising when describing it in the Results section. With respect to the interpretation of the effect, we refer the reader to the extended discussion of factors potentially influencing the direction of the association between pattern correlations and temporal relations (see our response to Comment 5). We believe that these changes have improved the presentation of our results because the manuscript now presents this effect in a way that is more accessible and makes clear that the direction of the effect is surprising. Please find the revised figure and the revised section of the manuscript below:

Page 7

Surprisingly, we observed a positive relationship between similarity changes and temporal distances in both the summary statistics (Figure 4B; $t_{27}=3.07$, $p=0.006$, $d=0.56$, 95% CI [0.18, 1.00]; $\alpha=0.025$, corrected for separate tests of events of the same and different sequences) and the mixed model approach (Figure 4CD; $\chi^2(1)=9.87$, $p=0.002$, Supplemental Figure 4CD, Supplemental Table 2). This effect was further characterized by higher pattern similarity for event pairs separated by longer temporal distances than for pairs separated by shorter intervals (Figure 4C, $t_{27}=2.48$, $p=0.020$, $d=0.64$, 95% CI [0.08, 0.87]). In contrast to our previous work²¹, where we observed negative correlations of pattern similarity and temporal distances, participants learned multiple sequences in this study.

Figure 4

Figure 4. Sequence representations in anterior hippocampus reflect constructed event times. **A.** The anterior hippocampus region of interest is displayed on the MNI template with voxels outside the field of view shown in lighter shades of gray. Color code denotes probability of a voxel to be included in the mask based on participant-specific ROIs (see Methods). **B.** The Z-values based on permutation tests of participant-specific linear models assessing the effect of virtual time on pattern similarity change for event pairs from the same sequence were significantly positive. **C.** To illustrate the effect shown in **B**, average pattern similarity change values are shown for same-sequence event pairs that are separated by low and high temporal distances based on a median split. **D.** Z-values show the relationship of the different time metrics to representational change based on participant-specific multiple regression analyses. Virtual time predicts pattern similarity change with event order and real time in the model as control predictors of no interest. **B-D.** Circles are individual participant data; boxplots show median and upper/lower quartile along with whiskers extending to most extreme data point within 1.5 interquartile ranges above/below the upper/lower quartile; black circle with error bars corresponds to mean \pm S.E.M.; distributions show probability density function of data points. ** $p < 0.01$; * $p < 0.05$

Comment 2

Related to point 1: It's a little confusing that "negative correlations between representational change and temporal distances" are actually in the prediction you would expect. Negative makes one think it's going to be the opposite of what would be expected.

Related to the previous comment, the reviewer here points out that the prediction of negative correlations between pattern similarity and distances might be unexpected for some readers. We understand this concern as the expectation of a negative correlation might seem uncommon. However, a negative correlation between pattern similarity and temporal distances is indeed what one would expect. In the temporal domain, this implies that representations of events that are separated by a small amount of time are more similar than those events that are separated by larger temporal intervals. As a different example, a similar pattern would be expected for map-like representations of space, where locations with low distances between them would share similar representations, whereas distant positions would have less similar representations. To make the interpretation

and expectation of negative correlations between pattern similarity and temporal distances clearer for the reader, we have expanded our description of these effects in our prior work in the introduction.

Please find the changed section of the manuscript below.

Page 2

Events closer in time elicited more similar activity patterns relative to events separated by longer intervals, resulting in negative correlations between pattern similarity and temporal distances^{21,27}. Within the entorhinal cortex, this effect was specific to the anterior-lateral subregion²⁷, consistent with the involvement of this area in precise temporal memory recall^{28,29}. Negative correlations between pattern similarity and distances are in line with sequence representations akin to cognitive maps of space – positions separated by low distances share similar representations, whereas positions with high distances between them are represented less similarly, i.e. pattern similarity scales with distance.

Comment 3

It was not entirely clear whether it was possible to infer any of the sequence or event structure from the images prior to the day learning task. E.g., do events from the same sequence have any greater visual or conceptual similarity (in terms of rooms, characters, objects, etc.)? This is not necessarily a problem given the pre vs. post design, but it would be useful to know.

The reviewer here asks whether task-relevant information about the sequences or the individual event times could be inferred from the stimuli. We do not believe that this was possible because of the way our stimuli and experiment were designed. The stimuli we created were devoid of contextual cues for the time of day on which they would occur. For example, no windows or clocks were visible that could indicate the time of day. Further, the event images used in the study were selected to be the most ambiguous with respect to the time of day when they might occur from a larger pool of images based on a pilot study. All event images featured the main character of the story. Crucially, we randomized across participants which events made up which sequence and which position in a sequence the events occupied. Thus, it was impossible to infer specific event times or sequence memberships from the stimuli. The randomization makes a systematic effect of visual or conceptual image content on our results very unlikely. Indeed, our searchlight analyses (Figure 7) revealed that the strongest effects were located in the hippocampus, whereas one would expect effects in visual cortices if they were driven by the visual similarity of the images. Further, as correctly pointed out by the reviewer, our analyses all focus on assessing representational change from the pre-learning baseline to the post-learning scan. As the visual and/or conceptual content of the images does not change, no systematic pattern similarity changes would be expected. We are thus confident that our effects are not driven by visual or conceptual similarity. We revised our manuscript to make better highlight that stimuli were randomly assigned to sequences and event times in the main text.

Page 4

Event images with minimal or no indication of time of day (Supplemental Figure 1) were randomly assigned to sequences and sequence positions for each participant. Thus it was impossible to infer specific event times or sequence memberships from the stimuli.

Comment 4

Would a similar interaction between temporal distance and sequence (within vs. across) be found if sequence position or actual clock time were used instead of virtual time? I don't know that it is a problem if the answer is "yes," but it would be interesting to know. Clearly subjects did learn the virtual times, but it is not clear that the surprising within-sequence temporal effects critically depend on virtual time.

The reviewer here asks whether not only virtual time, but also the two other time metrics (sequence order and real time) differentially relate to pattern similarity when comparing pairs of events belonging to the same sequence or two different sequences. We ran these analyses and observed significant interactions of sequence membership with order ($\chi^2(1)=9.98$, $p=0.002$) and real time ($\chi^2(1)=9.27$, $p=0.002$). Given that the three time metrics are related, similar interaction of each time metric in isolation can perhaps be expected. We next asked whether the interaction of virtual time and sequence membership would remain significant when including the other interaction terms in the mixed model. Indeed, virtual time was differentially related to pattern similarity for events from the same or different sequences, even when accounting for variance explained by the interactions of sequence membership with order and real time ($\chi^2(1)=8.57$, $p=0.003$). These results substantiate the interpretation that it is virtual time that underlies the differential representation of the temporal relations of the events from the same or different sequences. We have included these findings in the manuscript and report the mixed model details in Supplemental Table 8.

With respect to the last part of the comment, we believe that the data we present strongly argues for the within-sequence effect reflecting relations in virtual time. Specifically, even when competing for variance with order and real time, virtual time was a significant predictor pattern similarity change in the anterior hippocampus (summary statistics: $t_{27}=2.18$, $p=0.040$, $d=0.40$, 95% CI [0.02, 0.81], Figure 4D; mixed model: $\chi^2(1)=5.92$, $p=0.015$, Supplemental Figure 4EF, Supplemental Table 4). Further, virtual time significantly explained the residuals of linear models using order and real time as predictors (Supplemental Figure 5D; $t_{27}=2.23$, $p=0.034$, $d=0.41$, 95% CI [0.03, 0.82]). This indicates that systematic pattern similarity changes, which order and real time failed to account for, related to virtual temporal distances. In our view, these data back the interpretation that the within-sequence effect relates to virtual temporal distances between events.

The revised sections of the manuscript, which report the additional interaction analyses described above, read as follows:

Page 9

Similar interactions of sequence membership with order ($\chi^2(1)=9.98$, $p=0.002$) and real time ($\chi^2(1)=9.27$, $p=0.002$) were observed, but, crucially, the interaction of sequence membership and virtual time remained significant when including interactions of sequence membership with order and real time in the model ($\chi^2(1)=8.57$, $p=0.003$, Supplemental Table 8). Thus, the way knowledge about virtual temporal relations was represented in the hippocampus depended on whether events belonged to the same sequence or not.

Comment 5

Two potential accounts of the reduction in similarity for nearby, same-sequence events are offered. One is “These findings might support the view that the formation of associations between elements of a sequence may have altered how pattern similarity changes relate to temporal distances for same-sequence events.” I had difficulty understanding this argument. It would be helpful to flesh this out some more. The other account is a differentiation account. The current results do have a striking similarity to other examples of hippocampal differentiation of similar experiences. However, it is stated that “this interpretation does not directly account for the overall correlation of temporal distances and representational change.” I think this statement is true, but again could be fleshed out slightly more. I image the point is that a differentiation account does not explain the changes for across sequence representations?

The reviewer here refers to the interpretations of the same-sequence effect where temporal distances correlated positively with representational change, resulting in decreased pattern similarity for nearby events

relative to increases for distant events. We appreciate the opportunity to clarify the possible interpretations of this finding.

The idea that reliance on associative mnemonic strategies could underlie the surprising direction of the effect is grounded in the comparison of the experimental paradigm of the present study and our previous work (Deuker et al., eLife, 2016; Bellmund et al., eLife, 2019) as well as prior literature. Whereas participants learned when events occurred in only one sequence in our previous work, participants had to learn the times of events in four different sequences in the present study. Learning which events belong to the same sequence on top of learning the individual event times puts additional demands on the associative memory system. Thus, the two-fold nature of our sequence learning task (learning which events belong to the same sequence *and* learning the time of each event) might have altered how the hippocampus represented the temporal relations of events. Speculatively, forming associations between events from the same sequence, led to the difference in the direction of the correlation in a way where events at the on- and offset of the sequence were represented most similarly. Importantly, the same-sequence effect remains when controlling for this effect (Supplemental Figure 3, see also our response to the subsequent comment).

In line with these considerations, comparing previous studies from other groups also suggests that the use of associative strategies influences how hippocampal multi-voxel patterns relate to temporal memory. Jenkins and Ranganath (Hippocampus, 2016) asked participants to encode a stream of images from one category (i.e. one sequence) while undergoing fMRI. In a later memory test, participants judged the temporal order of two presented images. Hippocampal *pattern dissimilarity* at encoding was related to successful order memory recall. A potential mechanism could be the differentiation of contextual representations over time (Jenkins and Ranganath, Hippocampus, 2016; DuBrow & Davachi, Frontiers in Psychology, 2016). This interpretation is in line with the negative correlations between pattern similarity and temporal distances we previously observed (Deuker et al., eLife, 2016; Bellmund et al., eLife, 2019), where pattern similarity decreased with temporal distance. Conversely, the data reported by DuBrow & Davachi (Journal of Neuroscience, 2014) suggest that temporal memory can also be related to *increased pattern similarity*. Again, pattern similarity at encoding was assessed as a function of later order discrimination success. However, in this study participants encountered stimuli from two different visual categories (famous faces and objects) and were encouraged to use associative encoding strategies to remember the order of events. Indeed, participants were more accurate in remembering the order of stimuli if they were not separated by category switches, suggesting they associated the images presented between two switches to belong to one sequence. Hippocampal *pattern similarity was higher* for stimulus pairs whose order was later remembered correctly and additional analyses demonstrated the reinstatement of the stimuli separating the two probe stimuli (DuBrow & Davachi, Journal of Neuroscience 2014). These findings show that the relationship between hippocampal pattern similarity and temporal memory is multi-faceted and can depend on task demands and the encoding strategies participants employ. These theoretical considerations and previous findings underlie our interpretation of the positive correlations of pattern similarity and temporal distance for same-sequence events being due to the stronger reliance on associative encoding strategies compared to our previous work. We have clarified and expanded on this interpretation in the revised manuscript.

A second interpretation of the effect is that the hippocampus differentiates similar events, i.e. events that take place at similar times in the same sequence. As noted by the reviewer, this interpretation is in line with previous studies demonstrating hippocampal differentiation effects (Schlichting et al., Nature Communications, 2015;

Favila et al., Nature Communications, 2016; Chanales et al., Current Biology, 2017; Zeithamova et al., Journal of Neuroscience, 2018) and can serve to explain the pattern similarity decreases we observe for nearby events from the same sequence. We believe that this is a plausible explanation for this aspect of our results. However, a limitation of this interpretation is that it does not readily explain the relative increases in pattern similarity for temporally distant events from the same sequence. Likewise, hippocampal differentiation does not offer a straightforward explanation of the generalization effect we observed.

Following the reviewer's recommendation, we have expanded our considerations of the possible explanations of the same-sequence effect in the revised version of the manuscript. Please find the changed text below.

Page 14-15

The way temporal relations shaped hippocampal multi-voxel pattern similarity differed between pairs of events from the same and different sequences. We observed positive correlations between temporal distances and hippocampal representational change, which were characterized by relatively decreased pattern similarity for nearby compared to increased pattern similarity for more distant events from the same sequence. One possible explanation for the surprising direction of this effect could be that, compared to our previous work where participants encountered only one sequence²¹, participants relied more on associative encoding strategies when learning multiple sequences in the present experiment. Possibly, the need to link events belonging to the same sequence altered how pattern similarity changes relate to temporal distances for these same-sequence events. In line with this interpretation, prior work has shown that the relationship of hippocampal pattern similarity and temporal memory can depend on factors like the use of associative encoding strategies and the presence of event boundaries marking switches between sequences of images from the same category^{22,82,24}. Successful recency discrimination was associated with more similar hippocampal representations during encoding when participants were encouraged to use associative strategies to encode the order of image sequences from two alternating visual categories²². A different study found more dissimilar hippocampal representations for stimuli whose order was later remembered correctly²⁴. Thus, the formation of associations between same-sequence events could explain why correlations of pattern similarity change were, in contrast to our previous work²¹, positive. A second possible interpretation of this effect is based on observations that the hippocampus differentiates similar episodes^{47,83-86}. Hippocampal differentiation could explain the relative decrease of pattern similarity for temporally close events from the same sequence. However, the generalization across sequences does not directly follow from a differentiation account.

Comment 6

One potential account of the generalization results is that they may reflect some form of primacy or recency effects within the sequences. Namely, events that were presented at the beginning or end of the sequence might be better remembered. Or the first and last events might be more likely to trigger associative memory for the whole sequence—i.e., these events might be better memory cues for the sequence than events in the middle of the sequence. In either of these cases, the hippocampal response to 1st or last images would likely be different in a way that generalizes across sequences but without actually reflecting something about abstract time or position. In other words, you could observe the generalization effects even if the true nature of the hippocampal representation has nothing to do with abstract sequence position—it is just something that is correlated with sequence position. I do not see any obvious way to rule out this account. While I do view this as a limitation, I don't think this is a fatal flaw. Rather, I think this limitation should be acknowledged in the Discussion.

The reviewer here comments on the role of the events located at the first and last sequence positions. We agree that that these events might be of special importance when learning the sequences. Per definition, the

first and last events occur at the boundaries between sequences and might thus play a role in demarcating the sequences. We acknowledge that this is difficult to rule out entirely, but, in our view, both behavioral and neural data speak against the first and last event exclusively driving our effects.

If participants indeed formed stronger memories for the first and last events, we would expect that this would impact their memory for which sequence these events are associated with. However, the distribution of errors in the sorting task did not deviate from uniformity across sequence positions ($\chi^2=2.55$, $p=0.635$). This indicates that participants were no more or less likely to sort the first and last events to the correct sequence than the intermediate events. We have included this result in the manuscript and show the corresponding histogram below.

Histogram of the number of sorting errors as a function of sequence position. The distribution of errors did not deviate from uniformity ($\chi^2=2.55$, $p=0.635$).

In response to Reviewer 2, Comment 1 we conducted another, more fine-grained analysis of errors in the sorting task. We assessed swap errors, defined by participants swapping two events occupying the same sequence position between sequences in the sorting task. Again, one would predict these swap errors to occur more for the intermediate events, if participants had better memory for the sequence membership of the first and last events. However, the distribution of swap errors did not deviate from uniformity across sequence positions ($\chi^2=1.07$, $p=0.899$). This result is included in the revised manuscript and the corresponding histogram is shown in Supplementary Figure 3F.

With respect to hippocampal representational similarity, we have conducted new analyses, discussed above in response to Comment 4, which show that the interaction between sequence membership and virtual time also persists when including interaction terms for sequence membership and order as well as real time ($\chi^2(1)=8.57$, $p=0.003$). Furthermore, in response to Reviewer 2, Comment 3 we ran an additional analysis, which revealed that virtual time explains hippocampal representational change above and beyond the effects of order and real time also for events from different sequences (Summary statistics: $t_{26}=-2.62$, $p=0.015$, $d=-0.49$, 95% CI [-0.92, -0.10], mixed model: $\chi^2(1)=4.48$, $p=0.034$; one outlier excluded from these analyses due to a value more than 1.5 times the interquartile range from the mean). These findings highlight that an ordinal effect, which would also capture a strong impact of the first and last events, does not fully account for the hippocampal generalization effect we report.

The above findings do not provide evidence for the notion that the first and last events of the sequence drive the generalization effect. We have added these results to the revised manuscript. However, as also alluded to by the reviewer, it is difficult to entirely rule out that the first and last events of the sequences are particularly relevant for the task, which could potentially impact how the hippocampus generalizes across sequences. We now consider this possibility in the discussion section.

Please find below the revised sections of the manuscript.

Page 6

The distribution of sorting errors did not differ from uniformity across sequence positions ($\chi^2=2.55$, $p=0.635$).

Page 9

Virtual time was a significant predictor of hippocampal pattern similarity change for events from different sequences when competing for variance with order and real time (Supplemental Figure 6A-C; summary statistics: $t_{26}=-2.62$, $p=0.015$, $d=-0.49$, 95% CI [-0.92, -0.10], mixed model: $\chi^2(1)=4.48$, $p=0.034$, Supplemental Table 6; one outlier excluded).

Page 9

Similar interactions of sequence membership with order ($\chi^2(1)=9.98$, $p=0.002$) and real time ($\chi^2(1)=9.27$, $p=0.002$) were observed, but, crucially, the interaction of sequence membership and virtual time remained significant when including interactions of sequence membership with order and real time in the model ($\chi^2(1)=8.57$, $p=0.003$, Supplemental Table 8). Thus, the way knowledge about virtual temporal relations was represented in the hippocampus depended on whether events belonged to the same sequence or not.

Page 14

While it is possible that the first and last events of the sequences are particularly important to sequence processing, our data show that virtual time explained representational changes when competing for variance with order and real time also for events from different sequences. This makes it unlikely that the hippocampal generalization effect was driven exclusively by events at the first or last sequence position.

Supplemental Figure 3F

F. The distribution of swap errors over sequence positions did not deviate statistically from uniformity ($\chi^2(1)= 1.07$, $p=0.899$).

Supplemental Figure 6

Supplemental Figure 6. Virtual time predicts hippocampal pattern similarity change for events from different sequences. **A.** Z-values show the relationship of the different time metrics to representational change in the anterior hippocampus based on participant-specific multiple regression analyses for pairs of events from different sequences. Circles show participant-specific Z-values from summary statistics approach; boxplot shows median and upper/lower quartile along with whiskers extending to most extreme data point within 1.5 interquartile ranges above/below the upper/lower quartile; black circle with error bars corresponds to mean \pm S.E.M.; distribution shows probability density function of data points. **B, C.** Parameter estimates with 95% confidence intervals (**B**) and estimated marginal means (**C**) show the fixed effects of the three time metrics from the corresponding mixed model. * $p < 0.05$ after exclusion of one outlier excluded based on the boxplot criterion.

Comment 7

The within sequence hippocampal effects are reminiscent of a recent preprint from Sherman, DuBrow, Winawer, and Davachi <https://www.biorxiv.org/content/10.1101/2021.08.03.454949v1.full.pdf> which found that hippocampal pattern similarity within an event sequence was actually lower than across sequences. It might be worth acknowledging/citing this paper to add some conceptual support for the idea that the hippocampus differentiates events within a sequence.

The reviewer here refers to a recent preprint testing the role of event boundaries for duration judgments (Sherman et al., bioRxiv, 2021). Participants judged how long a square was presented on screen while undergoing fMRI. The color of the square was either constant across the entire trial or changed within the trial. When comparing the similarity of hippocampal multi-voxel patterns between the beginning and end of a trial, the authors found that pattern similarity was lower when the square did not change color compared to when it did change color. In their preprint, the authors offer two potential explanations for this finding. On the one hand, the color change might function as an event boundary that resets the neural population encoding the duration of an event. Assuming that neural codes diverge with time and that the same neural code is used at

the beginning of the trial and after the color change, this could result in more similar representations for the color change condition. On the other hand, the effect could be explained by a differentiation of patterns within an event. Given that there are two events in the color change condition, the one long event in the no-change condition could result in less similar patterns due to stronger separation over time. As pointed out by the reviewer, this latter explanation bears some resemblance with a differentiation of events in a sequence in our data.

We would like to thank the reviewer for pointing us towards this relevant preprint. However, we have decided not to reference it in our manuscript because the experiment described in the preprint by Sherman et al. (bioRxiv, 2021) does not disentangle the two interpretations and, in our view, also does not provide direct evidence for the separation interpretation. Rather, the pattern correlations in both conditions appear positive, possibly due to the temporal autocorrelation of the BOLD signal. We believe that this makes it difficult to interpret the data as a differentiation or separation effect. In contrast, prior work has interpreted decreases in pattern similarity through learning as evidence for hippocampal differentiation (e.g. Milivojevic et al., Current Biology, 2015; Chanales et al., Current Biology, 2017).

Comment 8

Related to the MDS reconstruction analysis:

(a) The reconstructions are really striking. But is this based on the change from pre to post learning or just based on the post learning results?

(b) Also, I initially assumed this was based on the raw pattern similarity values and was stunned by the consistency of the reconstructions across different sequences. But it seems the MDS was actually applied to modeled data. I did not entirely follow how this was done and, therefore, it is hard to ascertain whether the remarkable consistency of the reconstructions across sequences simply reflects that structure imposed by the model?

(c) It's a bit hard to reconcile the idea that the hippocampus 'learns' the sequence representation while also systematically showing decreased similarity for nearby events compared to far away events. These ideas seem to be at odds. Seeing the c-shape structure in the reconstructions really helps provide some intuition for this in that it seems that the start point and end point end up being close(r) together in representational space. Though, it is hard to tell (by eye) whether this reconstruction actually captures this effect. If you just compute the Euclidean distance between the reconstructed pairs, are temporally nearby events actually farther away (greater Euclidean distance) than temporally distant events?

We appreciate the opportunity to clarify how we used multidimensional scaling (MDS) to arrive at the two-dimensional visualization. In keeping with our main analyses, we focused on pattern similarity change, i.e. differences in pattern similarity computed by subtracting similarities in the pre-learning baseline scan from the post-learning scan. Because the resulting similarity matrices (one for each participant) can be noisy, we used the linear mixed model capturing the interaction effect of virtual temporal distances on hippocampal pattern similarity change to generate the input for MDS. Specifically, this is the mixed model including virtual temporal distances and a same/different sequence predictor as well as their interaction as fixed effects (Supplemental Figure 4IJ, Supplemental Table 7). As described in the results section, the significant interaction of this model describes the differential relationship of virtual temporal distances and pattern similarity for events from the same compared to events from different sequences. We derived a similarity matrix for all event pairs of our design given the virtual temporal distances between the constituting events and their sequence membership

from this model. This was done using the 'predict'-method on the resulting mixed model as implemented in the lme4 package for R. This similarity matrix was converted to a distance matrix and then subjected to MDS. To make this analysis procedure more accessible to the reader, we have expanded its description in the Methods and now show a schematic of the different steps in the new Supplemental Figure 6D.

The procedure described above is intended to reduce the noise present in hippocampal multi-voxel patterns by using a model-derived matrix as input for MDS. As noted by the Reviewer, this implies that the same relationship of virtual temporal distances and event similarity is assumed for all comparisons of events from different sequences (i.e. temporal distance should have the same effect when comparing events from sequences 1 and 2 and when comparing events from sequences 3 and 4). It is possible that this contributes to the parallel reconstruction of the different sequences, which we make explicit for the reader in the revised manuscript.

Regarding the last part of the comment, we agree that the exploratory visualization of the different sequences based on MDS helps to get an intuition for how the sequences might be arranged in a low-dimensional representational space. We contrasted the distances between event pairs in the MDS configuration based on a median split of input distances using a t-test for independent samples. Events separated by high input distances were separated by larger distances in the configuration resulting from MDS than those events separated by lower input distances ($t_{188}=9.35$, $p<0.001$, $d=1.35$, 95% CI [1.03, 1.67]). We also quantified the relationship between the input and MDS distances using a Spearman correlation and observed the expected positive correlation ($r=0.46$, $p<0.001$). Visual inspection of the underlying data points revealed that the association was not linear. Distances in the MDS configuration were lower than expected for event pairs separated by high input distances. This possibly reflects a limitation of projecting the data into a space with only two dimensions for visualization. We have added the analyses and these considerations to the manuscript.

Please find the revised sections of the manuscript below.

Page 9

To explore how event sequences may be arranged in a low-dimensional representational space to give rise to the effects described above, we generated a distance matrix from the mixed effects model fitted to hippocampal pattern similarity change and subjected it to non-metric multidimensional scaling (see Methods, Supplemental Figure 6D). The resulting configuration in two dimensions (Figure 5C), chosen for intuitive visualization, exhibited a c-shaped pattern for each sequence. Similar representational geometries have previously been described in parietal cortex⁶³⁻⁶⁵. Events occurring at similar virtual times occupy similar locations, in line with high pattern similarity for events from different sequences that are separated by low temporal distances. Thus, the generalization across sequences results in a comparable configuration for each sequence. While the observed configuration resulted in stress values significantly lower than those obtained in a permutation test (see Methods; $z=-3.5$, $p=0.001$, Supplemental Figure 6E), the high representational distances between temporally close events from the same sequence are not perfectly captured by the c-shaped arrangement (Supplemental Figure 6FG). More than the two dimensions chosen for visualization would likely better capture the complex representational structure of the sequences.

Page 23-24

Multidimensional Scaling

We aimed to explore how hippocampal event representations of the different sequences could be embedded in a low-dimensional representational space to give rise to the positive and negative correlations of pattern similarity change and temporal distances for same-sequence and different-sequence events, respectively. For each pair of events, we generated an expected similarity value (Supplemental Figure 6D)

using the fixed effects of the mixed model fitted to hippocampal pattern similarity that captures the interaction between virtual temporal distances and sequence membership (c.f. Figure 5, Supplemental Figure 4IJ, and Supplemental Table 7). Using the predict-method implemented in the lme4-package¹⁰⁹, we generated model-derived similarity values for all event pairs given their temporal distances and sequence membership. We chose this approach over the raw pattern similarity values to obtain less noisy estimates of the pairwise distances. Using the smacof-package¹¹¹, the model-predicted similarities were converted to distances and the resulting distance matrix (Supplemental Figure 6D) was subjected to non-metric multidimensional scaling using two dimensions. We chose two dimensions to be able to intuitively visualize the results. Because MDS is sensitive to starting values, we ran multidimensional scaling 1000 times with random initial configurations and visualized the resulting configuration with the lowest stress value. Basing this analysis on the model-derived similarities assumes the same relationship of virtual temporal distances for all event pairs from different sequences, but we would like to note that not all solutions we observed, in particular those with higher stress values, resulted in parallel configurations for the four sequences.

We tested the stress value of the resulting configuration against a surrogate distribution of stress values obtained from permuting the input distances on each of 1000 iterations. Using the mean and standard deviation of the resulting null distribution, we obtained a z-value as a test statistic and report the proportion of stress values in the null distribution that were equal to or smaller than the observed stress value (Supplemental Figure 6E). Additionally, we contrasted the distances between pairs of events in the resulting configuration between distances separated by high or low (median split) input distances using a t-test for independent samples (Supplemental Figure 6F). Using a Spearman correlation, we quantified the relationship of the input distances and the distances in the resulting configuration (Supplemental Figure 6G).

Supplemental Figure 6

D. A linear mixed model capturing the interaction effect of virtual temporal distances and sequence membership (Figure 5, Supplemental Figure 4IJ) was fitted to hippocampal representational change. An event-by-event similarity matrix was derived from the fixed effects of this model. Similarities were converted to distances and then used as input for multidimensional scaling (see Methods). **E.** The stress value observed in the MDS analysis (red line) was significantly smaller than the 5th percentile (black dashed line) of a

surrogate distribution of stress values obtained from shuffling the dissimilarities before running MDS in each of 1000 iterations. **F.** Pairs of events separated by a large distance in the input distance matrix were separated by a larger Euclidean distance in the resulting MDS configuration ($t_{188}=9.35$, $p<0.001$, $d=1.35$, 95% CI [1.03, 1.67]). *** $p < 0.001$. **G.** There was a significant Spearman correlation of input distances and MDS configuration distances ($r=0.46$, $p<0.001$), but visual inspection reveals a non-linear relationship where very high distances are systematically underestimated in the MDS configuration. This is likely because the data were projected onto only two dimensions for visualization. More dimensions would be needed to improve the fit of the MDS configuration and the input distance matrix. Distances are shown as ranks because non-metric MDS was used (high ranks for high distances).

Minor:

Comment 9

Line 270: “When events at the same sequence position, but in different sequences, took place late relative to an event...”. Is this supposed to say “relative to a SEQUENCE”? or maybe “relative to other events at the same sequence position”?

We would like to thank the reviewer for flagging that the wording was ambiguous here. We have revised the sentence to more clearly describe the generalization effect.

Page 13

The constructed virtual time of an event tended to be overestimated when the events occupying the same sequence position in the other sequences took place late relative to the event in question, and vice versa when the other events occurred relatively early.

Comment 10

Page 8: “we next assessed whether this effect was driven by the constructed event times beyond sequence order and real time. We thus included the two additional time metrics as control predictors in the model.” I initially interpreted this to mean that the behaviorally-constructed times from the post-test were used. But I think that what the authors might have meant is just the “virtual time.” If so, I would recommend strictly using “virtual time” to refer to the objective virtual clock given the potential to confuse the objective virtual time with the behaviorally-constructed virtual time. I do understand that virtual time HAS to be constructed, but, still, there is the potential for confusion.

We indeed refer to virtual times defined by our experimental design in this sentence. We followed the recommendation of the Reviewer to consistently use the phrase “virtual time” for this time metric throughout the manuscript. Please find the revised section of the manuscript below.

Page 8

Having established that hippocampal pattern similarity changes relate to temporal distances, we next assessed whether this effect was driven by virtual event times beyond sequence order and real time.

Reviewer #2

Bellmund and colleagues report a functional MRI study in which they sought to shed light on the type of temporal memory (i.e. inferred/constructed time, elapsed real time, temporal order) that anterior hippocampal and anterior lateral entorhinal representations encode in support of event sequence memory. Using a representational similarity approach together with a behavioural task that required participants to learn the temporal structure of four different event sequences that unfolded in relation to a 'virtual clock', it was found that anterior hippocampal representations contained information about

constructed time for both within and between sequence events. Specifically, events that were further apart in virtual time within the same sequence were associated with greater pattern similarity compared to events that were closer together. In contrast, events that occurred at comparable virtual times between sequences were represented more similarly than those that took place at different virtual times. This latter finding points towards the generalization of temporal relations across sequences and reflective of this, participants demonstrated a 'generalization bias' in their task responses, with the estimated virtual time of any given event being influenced by the time of occurrence of other events in other sequences in the same sequence position. In comparison to the anterior hippocampus, representations in the anterior lateral entorhinal cortex did not differentiate between same-sequence and across-sequence temporal relations, with events occurring close together in time being associated with greater pattern similarity.

There has been much interest in how the brain supports the temporal dimension of event sequence memory and as such, the current study is very timely, with the potential to be conceptually important. The behavioural paradigm is clever, providing a means to disentangle constructed time from elapsed time and event order, and broadly speaking, the experimental methods and data analyses are solid. Indeed, the authors are to be commended for their use of multiple analytical approaches (to provide converging findings) and the implementation of certain control analyses to rule out alternative explanations. Overall, I'm quite enthusiastic about this paper but have a number of issues that I would appreciate further clarity on.

We would like to thank the reviewer for their positive evaluation of our manuscript and the helpful suggestions for additional analyses. Specifically, we have followed the suggestion to explore systematic swap errors in the sorting task. We indeed found at least one swap error, i.e. a swap of events that occupy the same sequence position between sequences, in 12 of 14 participants who made errors in the sorting task. In total, swap errors accounted for around 57% of errors in the sorting task and were observed more frequently than expected from chance based on a permutation test. These findings provide intriguing additional evidence that participants' memory performance exhibits biases indicative of across-sequence generalization. This is paralleled by additional analyses of the fMRI data, which support the interpretation that the hippocampal across-sequence generalization effect is driven by event relations in virtual time rather than sequence order or real time. In sum, we believe that these new results substantially strengthened our manuscript. Please find our detailed responses to the individual comments below.

Major Comments

Comment 1

It is somewhat surprising that a handful of participants performed relatively poorly on the sorting task (in the 40 - 60 % correct range) and, in fact, there was quite a range in performance across participants. Were the poorest performers on the sorting task also those who produced the greatest absolute error on the timeline task? Related to this, it would be interesting to know whether there were any discernible patterns in the errors that participants made on the sorting task. Specifically, swap errors, in which participants mix-up events occupying the same sequence position across sequences (i.e. A1-B2-C1-D1-E1 & A2-B1-C2-D2-E2), may be a product of generalization across sequences and could conceivably be related to the generalization bias (e.g. greater generalization of temporal structure across sequences may lead to a greater number of swap errors and a greater generalization bias).

The reviewer here asks about the relationship between performance levels in the different memory tasks and whether participants made systematic errors in the sorting task. We explored whether the number of sorting

errors and absolute errors in the timeline task were correlated across participants, but observed no significant correlation (Spearman's $r=0.23$, $p=0.246$, Supplemental Figure 3C). However, despite there being variable performance in the timeline task, and, as noted by the reviewer, in the sorting task, this correlational analysis is limited by the fact that performance in the timeline task was at ceiling for many participants. Specifically, 14 of 28 participants made no errors in the sorting task, resulting in reduced between-subject variability. To further explore a possible relationship between the tasks, we split our sample based on whether participants made no or at least one sorting error and contrasted averaged absolute timeline errors using a t-test for independent samples. We observed no statistically significant difference when using a two-sided test ($t_{26}=1.79$, $p=0.085$, Supplemental Figure 3D). We would like to note that the numerical difference was in the expected direction and that our sample size does not yield large statistical power for the study of individual differences.

The reviewer further suggests investigating "swap errors", where, for events occupying the same position in their sequence, participants interchange which sequence the events belong to in the sorting task. We had not previously tested for systematic errors in this task and would like to thank the reviewer for proposing this intriguing analysis. We defined swap errors as errors in the sorting task, where two or more events from the same sequence position were incorrectly sorted to the respective other sequence(s). We observed that $57.5\pm 34.3\%$ of all sorting errors were swap errors, with 12 of 14 participants with sorting errors making at least one swap error (mean \pm S.D: 3.1 ± 2.1 swap errors, Supplemental Figure 3E). To test whether participants indeed systematically made swap errors we ran a permutation test where we introduced sorting errors for randomly selected events. For each of 10000 iterations, we generated a sample of surrogate sorting results with the number of randomly introduced sorting errors matching the number of errors made by the participants in our sample. We then quantified the proportion of swap errors across the sample, resulting in a chance distribution of how many swap errors would be expected if sorting errors were random. The proportion of swap errors in our data was larger than what would be expected from random sorting errors ($z=5.07$, $p<0.001$, Supplemental Figure 3G), indicating that participants indeed systematically interchanged events occupying the same sequence position.

The distribution of swap errors did not differ from uniformity across sequence positions (Supplemental Figure 3F, $\chi^2=1.07$, $p=0.899$). While the existence of swap errors is in line with participants generalizing across sequences also in the sorting task, leading to swaps between events occupying the same position in different sequences, we neither observed a statistically significant correlation across subjects between the number of swap errors and the generalization bias in the timeline task (Supplemental Figure 3H, Spearman $r=0.12$, $p=0.528$) nor a difference in the generalization bias between participants who did or did not make at least one swap error (Supplemental Figure 3I, $t_{26}=0.18$, $p=0.861$). Again, we would like to note that the statistical power of these across-subject analyses is limited by the fact that not all participants made sorting errors and the relatively small sample size for the investigation of individual differences.

We would like to thank the reviewer for the suggestion to take a closer look at the errors in the sorting task. We believe that the detection of swap errors adds another interesting behavioral finding to our manuscript that is in line with the notion that participants generalize across sequences. We have included the above analyses in the revised manuscript and show the results in the new Supplemental Figure 3. Please find the new sections below.

Page 6

We did not observe an across-subject relationship between the number of sorting errors and mean absolute errors in the timeline task (Supplemental Figure 3CD).

Page 13

We further explored whether participants made systematic errors in the sorting task that might point towards generalization across sequences. Specifically, we searched for swap errors where participants interchanged events between sequences that occupied the same sequence position. Indeed, $57.5\% \pm 34.3\%$ (mean \pm S.D) of sorting errors were swap errors and 12 of the 14 participants who made sorting errors also made swap errors (Supplemental Figure 3EF, mean \pm S.D of 3.1 ± 2.1 swap errors per participant with sorting errors). The proportion of swap errors in our sample was larger than expected from random sorting errors ($z=5.07$, $p<0.001$, Supplemental Figure 3G), indicating that participants systematically swapped events belonging to the same position between sequences. While we did not observe statistically significant relationships between swap errors and the generalization bias (Supplemental Figure 3HI), the prevalence of these errors is compatible with the view that participants generalized across events occupying the same sequence position.

Page 20

In an exploratory analysis, we searched for systematic errors in the sorting task. Specifically, we looked for swap errors where participants interchanged events occurring at the same position between two or more sequences. We used a χ^2 -test to assess whether the number of swap errors deviated from uniformity across sequence positions. To test whether participants made more swap errors than expected from chance we ran a permutation test where we introduced sorting errors for randomly selected events. For each of 10 000 iterations, we generated a surrogate sample of sorting results with the number of randomly introduced sorting errors matching the number of errors made by the different participants in our sample. We then quantified the proportion of swap errors across this surrogate sample. This resulted in a distribution of the proportion of swap errors that would be expected from random sorting errors. We assessed how many permutations yielded proportions of swap errors larger or equal to the proportion of swap errors observed in the fMRI sample to compute a p-value and further quantified a z-value as the difference between the observed swap error proportion and the mean of the chance distribution divided by the standard deviation of the chance distribution. We tested whether the number of swap errors was related to absolute errors in the timeline task (see below) using Spearman's correlation and a t-test for independent samples.

Supplemental Figure 3

Supplemental Figure 3. Memory performance. **A.** A permutation-based repeated measures ANOVA revealed a significant effect of sequence on mean absolute errors in the timeline task ($F_{3,81}=5.86$, $p<0.001$, post hoc contrasts: sequence 1 vs. 2: $t_{27}=3.38$, $p=0.001$, sequence 1 vs. 3: $t_{27}=-0.12$, $p=0.912$, sequence 1 vs. 4: $t_{27}=2.59$, $p=0.013$, sequence 2 vs. 3: $t_{27}=-2.92$, $p=0.001$, sequence 2 vs. 4: $t_{27}=-1.15$, $p=0.271$, sequence 3 vs. 4: $t_{27}=2.15$, $p=0.023$). * $p <$ Bonferroni-adjusted alpha-level of 0.008, corrected for 6 pairwise post hoc comparisons. **B.** Mean absolute timeline errors did not differ statistically between sequences with fast and slow clock speed ($t_{27}=-0.82$, $p=0.423$). **C.** The number of errors in the sorting task did not correlate with the mean absolute error in the timeline task across participants ($r=0.23$, $p=0.246$). **D.** Mean absolute errors in the timeline task were not statistically different between participants who made one or more errors (red) or no errors in the sorting task (green) in the sorting task (t-test for independent samples, $t_{26}=-1.79$, $p=0.085$). **E.** Histogram shows the number of swap errors for participants with (red) and without (green) errors in the sorting task. **F.** The distribution of swap errors over sequence positions did not deviate statistically from uniformity ($\chi^2(1)=1.07$, $p=0.899$). **G.** Histogram shows the null distribution of the proportion of swap errors expected under random sorting errors. The proportion of swap errors observed in our sample (red line) exceeded the 95th percentile of the null distribution (black line). **H.** The number of swap errors was not significantly correlated with the generalization bias (Spearman $r=0.12$, $p=0.528$). **I.** The generalization bias in the timeline task was not significantly different between participants who made one or more swap errors (red) or no swap errors (green) in the sorting task ($t_{26}=0.18$, $p=0.861$). **A, B, D, H.** Circles show individual participant values; boxplot shows median and upper/lower quartile along with whiskers extending to most extreme data point within 1.5 interquartile ranges above/below the upper/lower quartile; black circle with error bars corresponds to mean \pm S.E.M.; distribution shows probability density function of data points. **C, H.** Each circle shows data from one participant, grey line and shaded region indicate least squares line and confidence interval.

Comment 2

Was there a significant difference in absolute error on the timeline task between the different sequences? While eye-balling Figure 2B suggests that there wasn't, it may still be useful to demonstrate this statistically and show that clock speed had no impact on participants' ability to construct time.

The reviewer here asks whether the timeline errors differed between the four sequences and whether timeline errors were affected by the clock speed manipulation. We had not previously tested for these differences. A permutation-based repeated measures ANOVA with the mean absolute timeline error as the dependent variable and sequence as a within-subject factor revealed a significant effect of sequence ($F_{3,81}=5.86$, $p<0.001$). Pairwise post hoc tests showed this effect to be driven by lower errors for sequence 2 relative to sequences 1 and 3 (sequence 2 vs. 1: $t_{27}=-3.38$, $p=0.001$, sequence 2 vs. 3: $t_{27}=-2.92$, $p<0.001$, alpha-level Bonferroni adjusted for 6 post hoc comparisons). We next addressed the second part of the question by averaging absolute timeline errors for sequences with fast or slow clock speed. Importantly, we did not observe any differences between the two conditions (permutation-based paired t-test, $t_{27}=-0.82$, $p=0.418$). Because we did not find evidence for clock speed affecting the accuracy of inferred event times, we believe that the observed, non-systematic differences between sequences do not affect our key analyses, which focused on the comparison of different time metrics.

We have included the above analyses in the revised manuscript and show the results in the new Supplemental Figure 3AB. Please find the changed sections of the manuscript below.

Page 6

The accuracy of constructed virtual times differed between sequences ($F_{3,81}=5.86$, $p<0.001$), but not as a function of virtual clock speed ($t_{27}=-0.82$, $p=0.423$, Supplemental Figure 3AB).

Page 20

We quantified absolute errors across all events (Figure 2C) as well as separately for the five sequence positions (Figure 2D), the four sequences (Supplemental Figure 3A) and as a function of virtual clock speed (Supplemental Figure 3B).

Supplemental Figure 3AB

Supplemental Figure 3. Memory performance. **A.** A permutation-based repeated measures ANOVA revealed a significant effect of sequence on mean absolute errors in the timeline task ($F_{3,81}=5.86$, $p<0.001$, post hoc contrasts: sequence 1 vs. 2: $t_{27}=3.38$, $p=0.001$, sequence 1 vs. 3: $t_{27}=-0.12$, $p=0.912$, sequence 1 vs. 4: $t_{27}=2.59$, $p=0.013$, sequence 2 vs. 3: $t_{27}=-2.92$, $p=0.001$, sequence 2 vs. 4: $t_{27}=-1.15$, $p=0.271$, sequence 3 vs. 4: $t_{27}=2.15$, $p=0.023$). * $p <$ Bonferroni-adjusted alpha-level of 0.008, corrected for 6 pairwise post hoc comparisons. **B.** Mean absolute timeline errors did not differ statistically between sequences with fast and slow clock speed ($t_{27}=-0.82$, $p=0.423$).

Comment 3

When examining the representational similarity between events from different sequences in the anterior hippocampus (Supplemental Table 5), was an analysis conducted with real time and order included as additional control factors? The same applies for the examination of representational change in the anterior lateral entorhinal cortex (Supplemental Table 7). The findings of these analyses would be important to report.

The reviewer here asks whether virtual time explains pattern similarity changes beyond order and real time for the across-sequence effect in the hippocampus and the general effect in the entorhinal cortex. We would like to thank the reviewer for suggesting these interesting analyses, which we had not conducted before.

In the anterior hippocampus, the effect of virtual time was a significant predictor of pattern similarity change when competing for variance with order and real time in the across-sequence generalization analysis (Summary statistics: $t_{26}=-2.62$, $p=0.015$, $d=-0.49$, 95% CI [-0.92, -0.10], mixed model: $\chi^2(1)=4.48$, $p=0.034$; one outlier excluded from these analyses due to a value more than 1.5 times the interquartile range from the mean). As in the main analysis, we observed negative associations between virtual temporal distances and representational change. These findings provide further evidence that pattern similarity changes for events from different sequences were indeed reflective of virtual temporal distances.

We also conducted an analysis including virtual time, order and real time as predictors of pattern similarity change for comparisons both within and across sequences in the anterior-lateral entorhinal cortex. Virtual temporal distances did not significantly relate to entorhinal pattern similarity change when competing for variance with the other time metrics (summary statistics: $t_{27}=-0.7$, $p=0.495$, $d=-0.13$, 95% CI [-0.51, 0.25], mixed model: $\chi^2(1)=1.18$, $p=0.278$). One possible explanation for this is the lower signal-to-noise ratio in the entorhinal cortex, which we report in Supplemental Figure 8 of our manuscript. A second potential explanation is that the entorhinal cortex holds a representation that rather reflects the order of events or temporal distances based on elapsing real time since the start of the sequence. In particular, the latter possibility would be in line with the reports of signals varying with elapsing time in the rodent lateral entorhinal cortex (Tsao et al., Nature, 2018). However, these considerations remain on the level of speculation because, while real time significantly related to entorhinal pattern similarity changes ($t_{27}=-2.17$, $p=0.038$), real time was not significant when competing for variance with the other time metrics ($t_{27}=0.76$, $p=0.446$).

We have included the analysis of the hippocampal across-sequence effect with all time metrics in the model and the corresponding analysis in the entorhinal cortex in the revised manuscript. Please find the changed sections of the manuscript below.

Page 9

Virtual time was a significant predictor of hippocampal pattern similarity change for events from different sequences when competing for variance with order and real time (Supplemental Figure 6A-C; summary statistics: $t_{26}=-2.62$, $p=0.015$, $d=-0.49$, 95% CI [-0.92, -0.10], mixed model: $\chi^2(1)=4.48$, $p=0.034$, Supplemental Table 6; one outlier excluded).

Page 9

The relationship of virtual temporal distances and entorhinal pattern similarity change was not statistically significant when competing for variance with distances based on order and real time (Supplemental Figure 7B-D; summary statistics: $t_{27}=-0.7$, $p=0.495$, $d=-0.13$, 95% CI [-0.51, 0.25], mixed model: $\chi^2(1)=1.18$, $p=0.278$, Supplemental Table 10).

Supplemental Figure 6

Supplemental Figure 6. Virtual time predicts hippocampal pattern similarity change for events from different sequences. **A.** Z-values show the relationship of the different time metrics to representational change in the anterior hippocampus based on participant-specific multiple regression analyses for pairs of events from different sequences. Circles show participant-specific Z-values from summary statistics approach; boxplot shows median and upper/lower quartile along with whiskers extending to most extreme data point within 1.5 interquartile ranges above/below the upper/lower quartile; black circle with error bars corresponds to mean±S.E.M.; distribution shows probability density function of data points. **B, C.** Parameter estimates with 95% confidence intervals (**B**) and estimated marginal means (**C**) show the fixed effects of the three time metrics from the corresponding mixed model. * $p < 0.05$ after exclusion of one outlier excluded based on the boxplot criterion.

Supplemental Figure 7

B. Z-values show the relationship of the different time metrics to representational change in the anterior-lateral entorhinal cortex based on participant-specific multiple regression analyses. Analysis includes all pairs of events. **C, D.** Parameter estimates with 95% confidence intervals (**C**) and estimated marginal means (**D**) show the fixed effects of the three time metrics from the corresponding mixed model. **A, B.** Circles show participant-specific Z-values from summary statistics approach; boxplot shows median and upper/lower quartile along with whiskers extending to most extreme data point within 1.5 interquartile ranges above/below the upper/lower quartile; black circle with error bars corresponds to mean \pm S.E.M.; distribution shows probability density function of data points.

Comment 4

Given that generalization bias can be quantified for each trial, I'm curious as to whether the authors attempted an analysis whereby they examined trial-by-trial fluctuations in BOLD signal in relation to this measure (e.g. within a GLM with generalization bias as a regressor)? The observation of significant anterior HPC involvement in such an analysis would add further weight to the authors' conclusions.

The reviewer here suggests examining fluctuations of the BOLD signal during the picture viewing tasks. This is an interesting proposal that we had not previously considered because we optimized our task design for representational similarity analyses and obtained our behavioral read-out, which revealed the generalization bias, in a post-scan memory test and not during the picture viewing task where fMRI data were collected. In this memory test, each event image was arranged on a timeline once, whereas each image was shown multiple times in the fMRI picture viewing task. Behavioral analyses revealed a generalization bias quantified by the relationship between errors in the memory test and the relative time of the other events occupying the same sequence position. To relate this generalization bias to the picture viewing task, we used the relative time of other events at the same sequence position as a parametric modulator for specific event presentations.

To explore BOLD signal fluctuations related to the generalization bias we ran GLMs on the data from the post-learning picture viewing task for each participant. We included a regressor modeling the presentations of the twenty task-relevant event images as well as a parametric modulator for this regressor, which was defined based on the absolute difference in virtual time between a given event and the events at the same position in the other sequences. The activity pattern of a region that responds more strongly when the shown event is associated with a virtual time that deviates more from the average time of other events would be captured by this parametric modulation. We ran this GLM using FSL for each block of each participant and included additional regressors for the presentation of target pictures requiring a button press and for the motion parameters obtained during preprocessing. The z-values of the parametric modulation regressors were subjected to spatial smoothing (FWHM 3mm) and averaged across blocks within each participant. We used a permutation test (FSL Randomise, 5000 permutations, TFCE) for group-level statistical inference.

When restricting this analysis to our a priori ROIs, the anterior hippocampus and the anterior-lateral entorhinal cortex, we observed a cluster of voxels in the right anterior hippocampus that exhibited the described parametric modulation at uncorrected levels (peak voxel at MNI x=22, y=-13, z=-16, t=3.93, $p_{sv_uncorrected} < 0.001$, see Figure below). However, this cluster was not statistically significant after correction for multiple comparisons within our a priori ROIs ($p_{svc}=0.173$). When including all voxels in our field of view in the analysis, no voxels reached a significance threshold of $p < 0.001$ (uncorrected).

These results provide some evidence for the BOLD fluctuations in line with the generalization bias, but we believe they are too preliminary to be included in the manuscript. Our experimental design was optimized for

Results of the parametric modulation analysis. The response of a cluster of voxels in the right anterior hippocampus (peak voxel MNI $x=22$, $y=-13$, $z=-16$, $t=3.93$, $p_{sv_uncorrected}<0.001$) to event images was modulated by the relative time of other events at the same sequence position. The statistical image is masked for the anterior hippocampus and anterior-lateral entorhinal cortex and thresholded at $p_{uncorrected}<0.05$ in these regions.

the analysis of representational changes from the pre-learning baseline to the post-learning scan. Keeping the pre-learning and post-learning scans as similar as possible precluded us from including a behavioral test of participants' memory for the event times during the picture viewing task. Possibly, the parametric modulation of hippocampal activity by the relative time of other events would have been stronger if participants had to behaviorally respond to each event image.

Comment 5

It would be useful if the authors could clarify why they used NMDS to visualize the relationship between the different events, sequences and virtual time, as opposed to MDS, and how they explored/determined the optimal number of dimensions to account for their data. Moreover, to further quantify the output of the NMDS analysis, one could consider using k-means clustering to examine the clustering of the different events across the different sequences.

We appreciate the opportunity to clarify how we approached the multidimensional scaling analysis (MDS). With this method, we hoped to gain insight into how the events could be arranged in a low-dimensional representational space given that the RSA results demonstrate representations of temporal relations for event pairs from the same and from different sequences. We thus chose two dimensions to be able to intuitively visualize the outcome of this exploratory analysis. We have made this explicit in the revised manuscript.

In our manuscript we show the configuration resulting from non-metric MDS. We also explored how these results change when using metric MDS. The configurations from the two methods are shown below. Notably, the metric solution is somewhat similar to the one from non-metric MDS in that the events at similar sequence positions are located at similar positions, resulting in largely parallel lines when connecting events from the same sequence. The results from permutation tests on the two solutions are the reason we chose to include the non-metric version in our manuscript. Specifically, we used permutation tests as implemented in the *smacof* package for R to assess the goodness-of-fits for the two variants. On each of 1000 iterations, the input dissimilarities were permuted before running metric or non-metric MDS. This procedure yields a null distribution of stress values against which the originally observed stress value can be compared. The permutation test revealed that the solution of non-metric MDS gave a stress value lower than expected under the null distribution ($z=-3.5$, $p=0.001$). The results of this permutation test are shown in the new Supplemental Figure 6E of the manuscript. For metric MDS, the stress value was not lower than expected from the corresponding null distribution ($z=3.3$, $p=0.999$). These results indicate that the non-metric, but not the metric, MDS solution

captured the underlying dissimilarities between event pairs, which is why we reported the non-metric version in our manuscript.

Results of non-metric (A) and metric (B) MDS.

The reviewer further makes the interesting suggestion to investigate the resulting configuration further. In response to a similar comment by Reviewer 1 (Comment 8) we assessed how closely the distances between event pairs were maintained in the non-metric MDS solution. When assessing distances between events in the MDS configuration we observed that events separated by large input distances were separated by larger distances in the MDS configuration compared to events with low input distances (median split, $t_{188}=9.35$, $p<0.001$, $d=1.35$, 95% CI [1.03, 1.67], new Supplemental Figure 6F). Further, there was a significant Spearman correlation between input distances and distances in the MDS configuration ($r=0.46$, $p<0.001$, new Supplemental Figure 6G). Visual inspection of the underlying data point revealed that large input distances were systematically underestimated in the two-dimensional MDS solution. Returning to the question about the number of dimensions used for MDS, we believe that this is likely due to the complex nature of the hippocampal representation of the sequences, which was characterized by positive correlations of pattern similarity and virtual temporal distances for same-sequence events and negative correlations for events from different sequences. While our non-metric MDS solution results in a fit that is better than chance, projections into higher-dimensional spaces would be needed to fully capture the structure of the underlying representation.

In the revised manuscript, we have made explicit that we chose two dimensions for the MDS analysis for intuitive visualization. We further added the additional analyses of the resulting configuration and the consideration that a larger number of dimensions would be necessary to fully capture the representational structure.

Page 9

To explore how event sequences may be arranged in a low-dimensional representational space to give rise to the effects described above, we generated a distance matrix from the mixed effects model fitted to hippocampal pattern similarity change and subjected it to non-metric multidimensional scaling (see Methods, Supplemental Figure 6D). The resulting configuration in two dimensions (Figure 5C), chosen for intuitive visualization, exhibited a c-shaped pattern for each sequence. Similar representational geometries have previously been described in parietal cortex⁶³⁻⁶⁵. Events occurring at similar virtual times occupy

similar locations, in line with high pattern similarity for events from different sequences that are separated by low temporal distances. Thus, the generalization across sequences results in a comparable configuration for each sequence. While the observed configuration resulted in stress values significantly lower than those obtained in a permutation test (see Methods; $z=-3.5$, $p=0.001$, Supplemental Figure 6E), the high representational distances between temporally close events from the same sequence are not perfectly captured by the c-shaped arrangement (Supplemental Figure 6FG). More than the two dimensions chosen for visualization would likely better capture the complex representational structure of the sequences.

Page 23

We chose two dimensions to be able to intuitively visualize the results.

Supplemental Figure 6

D. A linear mixed model capturing the interaction effect of virtual temporal distances and sequence membership (Figure 5, Supplemental Figure 4IJ) was fitted to hippocampal representational change. An event-by-event similarity matrix was derived from the fixed effects of this model. Similarities were converted distances and then used as input for multidimensional scaling (see Methods). **E.** The stress value observed in the MDS analysis (red line) was significantly smaller than the 5th percentile (black dashed line) of a surrogate distribution of stress values obtained from shuffling the dissimilarities before running MDS in each of 1000 iterations. **F.** Pairs of events separated by a large distance in the input distance matrix were separated by a larger Euclidean distance in the resulting MDS configuration ($t_{188}=9.35$, $p<0.001$, $d=1.35$, 95% CI [1.03, 1.67]). *** $p<0.001$. **G.** There was a significant Spearman correlation of input distances and MDS configuration distances ($r=0.46$, $p<0.001$), but visual inspection reveals a non-linear relationship where very high distances are systematically underestimated in the MDS configuration. This is likely because the data were projected onto only two dimensions for visualization. More dimensions would be needed to improve the fit of the MDS configuration and the input distance matrix. Distances are shown as ranks because non-metric MDS was used (high ranks for high distances). *** $p<0.001$

Minor:

Comment 6

In Figures 5D and 6E, it would be useful if the legend could state how temporal distances were classified as 'high' vs. 'low'

We agree that this is a relevant methodological detail and have included it in the captions of the revised figures.

Figure 4C

C. To illustrate the effect shown in **B**, average pattern similarity change values are shown for same-sequence event pairs that are separated by low and high temporal distances based on a median split.

Figure 5B

B. To illustrate the effect shown in **A**, average pattern similarity change values are shown for across-sequence event pairs that are separated by low and high temporal distances based on a median split.

Figure 6C

C. To illustrate the effect in **B**, raw pattern similarity change in the anterior-lateral entorhinal cortex was averaged for events separated by low and high temporal distances based on a median split.

Comment 7

It would be helpful for r and p values to be reported in the correlation figures (e.g. Figure 8B, Sup. Figure8).

We would like to thank the reviewer for the suggestion to improve our visualization of the single-subject data of the generalization bias. We have incorporated the Pearson correlation values for the individual scatter plots in Figure 8B and Supplemental Figure 10. Because we test for statistical significance on the group level and not for each individual participant, we have not included p -values in the plots.

Please find the changed Figures below.

Figure 8B

B. The scatterplot illustrates the generalization bias for an example participant. Each circle corresponds to one event and the regression line highlights the relationship between the relative time of other events and the errors in constructed event times. The example participant was chosen to have a median-strength generalization bias. See Supplemental Figure 8 for the entire sample. Correlation coefficient is based on Pearson correlation.

Supplemental Figure 10 (for space reasons we only show panel A here)

Supplemental Figure 10. Generalization bias in individual participants. A, B. Each panel shows the data from one participant. Each circle corresponds to one event. The x-axis indicates the average relative time of the events occupying the same sequence position in other sequences. The y-axis shows the signed error of constructed event times as measured in the timeline task. The regression line and its confidence interval are overlaid in red. Positive slopes of the regression line indicate that constructed event times are biased by the average time of events in the other sequences. Correlation coefficients are based on Pearson correlation. **A** shows data from the main sample; **B** from the replication sample.

Comment 8

To test whether within- and across-sequence representations overlap, ROIs were defined using a $p < 0.01$ uncorrected threshold. How was this threshold chosen and do the findings change dramatically if a threshold of $p < 0.001$ uncorrected is adopted?

The reviewer here asks about the statistical threshold used to define the region of interest for the analysis of representational change in the cluster observed in the searchlight analysis looking for a relationship between virtual temporal distance and pattern similarity. We had chosen the threshold of $p < 0.01$ to obtain a cluster with a larger number of voxels to obtain stable pattern similarity estimates. In response to this comment, we re-ran this analysis with the suggested threshold of $p < 0.001$. We observed no statistical difference between the results obtained with the different thresholds ($t_{27} = -0.95$, $p = 0.338$, test against 0 using the ROI resulting from the $p < 0.001$ threshold: $t_{27} = -1.98$, $p = 0.056$).

We have added these details to the revised manuscript.

Page 23

The results observed using a threshold of $p < 0.001$ were not statistically different from those obtained with a threshold of $p < 0.01$ ($t_{27} = -0.95$, $p = 0.338$; test against 0 using the ROI resulting from the $p < 0.001$ threshold: $t_{27} = -1.98$, $p = 0.056$).

Reviewer #3

In this manuscript, Bellmund and colleagues interrogate the creation of temporal sequences in the anterior hippocampus and entorhinal cortex. They pursue this through the creation of a well designed experimental

paradigm wherein subjects watch distinct event sequences from different imaginary days in the life of a family. The authors attempt to dissociate 'virtual time' in the context of the viewed events and the absolute passage of time, doing so by providing participants with intermittent clock time cues indicating virtual time. The speed of these virtual clocks was manipulated such that the passage of virtual time and absolute time varied across simulated days. A pre/post still-frame viewing design is used to ascertain the differences in representational similarity for images that were presented both within the same sequence and across different sequences. Participants structured their estimates of the temporal structure of events strongly in accordance with virtual time, as opposed to mere temporal order or the real passage of time. This was mirrored by RSA results in the anterior hippocampus and anterior-lateral entorhinal cortex. Furthermore, the authors report that more temporally distant events within the same sequence become similar in the anterior hippocampus, suggesting that this may be due to beginning and end events being more strongly related. Temporally similar events across sequences, on the other hand, become more similar suggesting a realignment of episodes along a common axis. Conversely, the authors report that the entorhinal cortex featured only generalization of sequence structure rather than sensitivity to within versus across sequence information, such as they observed in the hippocampus. Finally, multidimensional scaling approaches were used to visualize low-dimensional embedding of the sequences of events, which I found to be a nice addition over the prior version of the manuscript.

My overall impression is that this is a very well-designed experiment and a fairly novel approach to studying temporal coding in the human medial temporal lobe. The notion of dissociating virtual time from real time is especially interesting (though, as I note below, I have some questions about the extent to which this is compellingly done). However, there are issues with conceptual framing of the results, and clarity about the way this work fits into and adds to the various corners of the literature that the authors cite. These points are listed below.

Lastly, I will note that I reviewed a prior version of this manuscript for a different journal. Though the authors addressed a few of the concerns I previously voiced, the authors did not really contend with the majority of the points raised from my prior review. Thus, I will largely reiterate those points, and note a few places where progress was made in my view over the previous version.

We would like to thank the Reviewer for taking the time to again review our manuscript and for the constructive criticism. We believe that addressing the comments has helped us to more clearly convey the main findings of our experiment in the restructured discussion and to better position our manuscript with respect to existing literature. Further, we have expanded our rationale for using two parallel analysis streams, which provide converging evidence, throughout our manuscript. In an effort to make the main figures more accessible to the reader, we have moved the panels showing the mixed model results to a supplemental figure. Overall, we believe that addressing these comments has improved our manuscript.

Major Points:

Comment 1

In my prior review, I noted that the framing of the paper was a bit opaque in terms of the study's contributions and scope with regards to the broader literature. Unfortunately, the authors do not seem to have changed very much at all from the previous version to address this concern. I was previously and still am struggling a bit with understanding the major contributions this study makes to changing or solidifying our understanding of the temporal organization of memory. A basic take-away from the paper is that the hippocampus and entorhinal cortex encode temporal sequences, which has been well-established (e.g.,

work from the authors' own group, as well as work by the Eichenbaum, Davachi, Ranganath, Howard, Fortin, and Kahana labs, among many others). Beyond basic sequence coding, the authors show that this temporal coding appears to align with virtual time rather than absolute time. This accords with recent work from Shimbo et al (Science Advances, 2021), and is novel in terms of human behavior and neural signals in the human brain. However, the framing of the manuscript is a bit scattered and does not effectively communicate this. I previously noted that the Discussion was particularly problematic in this regard. In this newer version, several minor changes seem to have been made which have made the Discussion read better. However, I still believe that the basic issue of too many tenuous links to what appear to be 'hot' research topics is detracting from rather than improving this manuscript.

The Reviewer here comments on the positioning of our manuscript in the literature and the necessity to more clearly communicate the contributions of this study. In our view, our study makes several contributions that advance our understanding of how the hippocampus represents sequences of events. We have restructured our discussion section along the following lines to more clearly communicate these findings and their relevance:

First, we show that the similarity of event representations is explained by the relations of events in virtual time. Participants constructed these event times from virtual time cues and their experience of elapsing real time. While virtual time could not be inferred without tracking real time (as discussed in more detail in our response to Comment 3), our results show that the relationship of hippocampal pattern similarity and virtual temporal relations extended beyond the contributions of real time and sequence order, demonstrating the impact of mnemonic construction on hippocampal representations of a relational structure.

Second, we demonstrate that the hippocampus forms an integrated representation of the different sequences such that temporal relations are generalized across sequences. This finding provides novel evidence for the way the hippocampus represents multiple sequences. In particular, it suggests that, in our task, it arranges them along one underlying dimension, resulting in the across-sequence generalization of temporal relations. This generalization of relational knowledge across sequences is in line with the role of the hippocampus in memory integration, inferential reasoning and generalization. Speculatively, this generalization effect could be related to a recent observation made in an experiment investigating hippocampal sequence representations in rodents (Sun et al., Nature Neuroscience, 2020). In this study, mice were trained to run a number of laps on a maze to obtain rewards and the authors report the emergence of lap-specific firing patterns in the hippocampus. Intriguingly, these lap-specific representations generalized across sequences of laps on geometrically distinct mazes. Consistent with the cross-sequence generalization we report, these findings suggest that the hippocampus can form transferable representations of temporal information.

Third, we show that structural knowledge influences the mnemonic construction of individual event times by detecting biases in behavior. This connects the two findings above by demonstrating that mnemonic construction and generalized, structural knowledge interact. This finding is in line with structural knowledge providing a scaffold for mnemonic construction of specific events.

The Reviewer further points to the interesting report of temporal scaling of time cell representations in the rodent hippocampus (Shimbo et al., Science Advances, 2021). Importantly, as in other reports of time cells, the data is assessed as a function of real elapsing time. The authors demonstrate that time cell representations compress or stretch depending on how the duration of a delay, during which the animal runs on a treadmill, is changed. This shows a remarkable flexibility of the time-cell representation and we do believe that this finding could be related to our observations. A difference we would like to note is that in the study by Shimbo et al.

time cells scaled to variations of real time intervals, whereas we show that the hippocampal representation covaries with mnemonically constructed times of events in sequences that had similar real time durations. Thus, the hippocampal representation scaled to this psychological, virtual time rather than changes in real time. The memory-based construction of a temporal representation would be difficult to study in animal models of temporal coding. In the revised discussion, we have included the possibility that our effects are related to the scaling of temporal representations observed in the rodent hippocampus.

Overall, we have followed the Reviewer's suggestion to more clearly communicate the key findings of our study and to provide a more focused account of their relationship to the existing literature. Please find the revised sections of the manuscript below.

Page 13-14

Our findings show that hippocampal event representations change through learning to reflect temporal relations based on mnemonically constructed event times. Converging region of interest and searchlight analyses demonstrate that, on the one hand, the hippocampus forms specific representations of temporal relations of the events in a sequence that mirror constructed event times beyond the effects of order and real time. On the other hand, temporal relations are generalized across sequences using a different representational format. In contrast, the similarity of event representations in the anterior-lateral entorhinal cortex scaled with temporal distances for events irrespective of sequence membership. The behavioral data demonstrate that the construction of specific event times is biased by structural knowledge abstracted from different sequences.

In our paradigm, participants mentally constructed the times of events relative to a hidden virtual clock. To do so, they needed to combine their experience of passing real time with infrequent cues about the current virtual time. Thus, real time was critical for the successful construction of event times, despite not being cued explicitly. Participants' responses in a memory test and the similarity structure of hippocampal multi-voxel patterns were explained by virtual event times beyond the effects of real time and sequence order, showing that sequence representations reflect mnemonically constructed time. Recent work demonstrated the scaling of time cell representations to different real time intervals in the rodent hippocampus⁶⁷. Temporal scaling of hippocampal representations could potentially underlie our observation that temporal distances in virtual time are related to the similarity of event representations even when accounting for the effects of real time and order. This finding highlights that the anterior hippocampus maps relational knowledge derived from mnemonic constructions.

The hippocampus constructed an integrated representation that generalized temporal relations across sequences. Multi-voxel patterns of events taking place at similar virtual times, but in different sequences, were more similar than those of events occurring at different points in time. Thus, representations of events from different sequences changed systematically to reflect generalized temporal distances. Speculatively, this effect could be related to the observation that, in mice trained to run a number of laps on a maze to obtain rewards, lap-specific firing patterns in the hippocampus generalize across sequences of laps on geometrically distinct mazes⁵⁴. While it is possible that the first and last events of the sequences are particularly important to sequence processing, our data show that virtual time explained representational changes when competing for variance with order and real time also for events from different sequences. This makes it unlikely that the hippocampal generalization effect was driven exclusively by events at the first or last sequence position. The generalization of temporal distances across sequences in the hippocampus is in line with the contribution of constructive mnemonic processes to flexible cognition via the recombination of elements across experiences and statistical learning^{13,40,43,46,48,49,68,69}. More generally, it is consistent with the role of the hippocampus in forming cognitive maps of relational structures and in generalizing structural knowledge to novel situations^{12,38,51,53,57,70,71}.

Comment 2

Following from the above, at a more nuanced level, the authors show that the hippocampus shows higher pattern similarity for same-sequence items which are farther apart than items that are closer together in time. The opposite pattern is found for different-sequence items. The authors note the following: “In contrast to our previous work, participants studied multiple sequences. They might have formed strong associations of same-sequence events on top of inferring each event’s virtual time, potentially altering how temporal distances affected hippocampal pattern similarity.” This is seemingly rephrased from the previous submission, but I am having difficulty understanding this point and do not find it to be an improvement. Given that the opposing pattern of results for same versus different sequences in the hippocampus is a key feature in the data, I think that a clearer discussion of this effect’s directionality (especially given that it is perhaps counterintuitive) is necessary.

The reviewer here asks about the interpretation of the positive relationship between temporal distances and hippocampal pattern similarity changes for same-sequence events. We appreciate the opportunity to clarify our views on the direction of this effect. In particular, we will compare the experimental paradigm used here to our previous work and discuss our findings in the light of previous findings on the role of associative mnemonic strategies on the relationship of hippocampal pattern similarity and temporal memory.

The present paradigm adds additional associative memory demands compared to the design of our previous experiment (Deuker et al., eLife, 2016; Bellmund et al., eLife, 2019). In our prior work, participants learned the temporal relationships of events that occurred in one sequence. In the present manuscript, participants need to learn which events belong to the same sequence in addition to learning the temporal structure of the sequences. To distinguish the different sequences, participants might use specific associative strategies, e.g. they might have formed links between same-sequence events. This might have altered how the hippocampus represented temporal relations *within* a sequence, resulting in a representational scheme where events at the on- and offset of the sequence were represented most similarly. However, please note that the effect of virtual temporal distances does not exclusively drive the same-sequence effect, which remains significant when including a predictor accounting for variance that is explained by pairs of events located at the first and last sequence position (Supplemental Figure 3).

Consistent with the notion that enhanced reliance on associative memory can alter how pattern similarity relates to temporal distances, previous studies from other groups point towards an impact of associative encoding strategies on the relationship between hippocampal pattern similarity and temporal memory. In one study (Jenkins & Ranganath, Hippocampus, 2016), participants encoded a sequence of images from one category while fMRI data were recorded. Subsequently, they judged the order in which pairs of stimuli had been presented. Successful temporal memory recall was associated with *less similar hippocampal patterns* during encoding. This is in line with the negative correlations between pattern similarity and temporal distances we observed in the across-sequence analysis and also reported previously (Deuker et al., eLife, 2016; Bellmund et al., eLife, 2019) and could be due to a differentiation of temporal context representations (Jenkins and Ranganath, Hippocampus, 2016; DuBrow & Davachi, Frontiers in Psychology, 2016). A different study however, observed successful order memory recall to be associated with *increased pattern similarity* in the hippocampus (DuBrow & Davachi, Journal of Neuroscience, 2014). Here, participants studied sequences of images from two different visual categories with a number of same-category images appearing in a row before the visual category switched. Importantly, participants were encouraged to use an associative encoding strategy and their less accurate order discrimination between stimulus pairs separated by category switches suggests that they

indeed remembered successively presented images from the same category as a sequence (DuBrow & Davachi, *Journal of Neuroscience*, 2014). These findings suggest that the relationship between hippocampal pattern similarity and temporal memory is modulated by 1) the number of sequences encoded by participants and 2) the mnemonic strategies they employ. This could underlie the positive correlations between temporal distances and hippocampal pattern similarity that we observe in the present study. We now have significantly expanded our discussion of this finding in the revised manuscript to make this interpretation clearer.

A second potential interpretation of the same-sequence effect is that it might go back to the differentiation of (temporally) similar events in the hippocampus. The positive correlation of temporal distances is characterized by lower pattern similarity for temporally close events relative to temporally far events (c.f. Figure 4C). This could be due to the hippocampus driving representations of events that take place at similar times in the same sequence apart. Indeed, prior work has demonstrated that the hippocampus differentiates similar episodes (Schlichting et al., *Nature Communications*, 2015; Favila et al., *Nature Communications*, 2016; Chanales et al., *Current Biology*, 2017; Zeithamova et al., *Journal of Neuroscience*, 2018). In our view, this is a plausible explanation for the relative decrease in similarity for nearby same-sequence events that contributes to the overall positive correlation of pattern similarity and distance. However, a differentiation account does not readily provide an explanation for the relative increases in pattern similarity for more distant same-sequence events or the generalization across sequences that we describe.

We have revised the paragraph in which the same-sequence effect is discussed to offer clearer and more comprehensive interpretations of this finding. Please find below the revised paragraph.

Page 14-15

The way temporal relations shaped hippocampal multi-voxel pattern similarity differed between pairs of events from the same and different sequences. We observed positive correlations between temporal distances and hippocampal representational change, which were characterized by relatively decreased pattern similarity for nearby compared to increased pattern similarity for more distant events from the same sequence. One possible explanation for the surprising direction of this effect could be that, compared to our previous work where participants encountered only one sequence²¹, participants relied more on associative encoding strategies when learning multiple sequences in the present experiment. Possibly, the need to link events belonging to the same sequence altered how pattern similarity changes relate to temporal distances for these same-sequence events. In line with this interpretation, prior work has shown that the relationship of hippocampal pattern similarity and temporal memory can depend on factors like the use of associative encoding strategies and the presence of event boundaries marking switches between sequences of images from the same category^{22,82,24}. Successful recency discrimination was associated with more similar hippocampal representations during encoding when participants were encouraged to use associative strategies to encode the order of image sequences from two alternating visual categories²². A different study found more dissimilar hippocampal representations for stimuli whose order was later remembered correctly²⁴. Thus, the formation of associations between same-sequence events could explain why correlations of pattern similarity change were, in contrast to our previous work²¹, positive. A second possible interpretation of this effect is based on observations that the hippocampus differentiates similar episodes^{47,83-86}. Hippocampal differentiation could explain the relative decrease of pattern similarity for temporally close events from the same sequence. However, the generalization across sequences does not directly follow from a differentiation account.

Comment 3

From my prior review, not addressed: An important aspect of this design and of the results is that behavioral and neural data indicate that participants are encoding information on the basis of virtual time,

rather than real time. Given the design of the experiment and stimuli, I am unsure if this should be at all surprising. Participants received any external cue whatsoever about virtual time, compared to none about real time. In fact, aside from aiding in one's understanding of how much time has elapsed between virtual clock cues, it does not seem that real time is at all relevant to participants' ability to engage in the task. While I have no doubt that participants are tracking the virtual timing of these events, or that they are doing so more strongly than they are absolute time, I am not sure how meaningful this comparison really is in the context of the experiment. While the authors did vary virtual clock speed in an attempt at dissociating real from virtual time, one can still reasonably argue that real time is uncued and simply not relevant to completing the task, rendering real time relatively uninformative, and a virtual versus real time comparison a strawman. Moreover, one could reasonably argue that participants' understanding of virtual time involves a combination of virtual + real time, further obscuring meaningful comparisons with real time only. I think the manuscript needs to address this issue.

The Reviewer asks an important question about the different time metrics that can be used to describe the temporal structure of the sequences, in particular about the role of real time in our experiment and its relationship to virtual time. The Reviewer correctly points out that participants did not receive cues about real, but only about virtual time. Yet, we would like to emphasize that real time is critical to inferring the virtual times of events, the central objective of our behavioral paradigm. The task cannot be solved without tracking the real time that elapses between virtual time cues to infer the virtual time of events. We have made more explicit in the revised manuscript that real time was uncued, but required to infer virtual times.

Despite the fact that the construction of virtual event times was necessarily based on real time, we believe that our analyses of the different time metrics are informative and provide novel insights into hippocampal sequence representations. As real time was critical for successful task performance, it was a relevant variable for participants to track. There is a substantial body of evidence implicating the hippocampus in representations of real time. This includes for example the discovery of so-called time cells, typically described as neurons firing at specific time points during a real time interval (e.g., Pastalkova et al., *Science*, 2008; MacDonald et al., *Neuron*, 2011). Further, we and others have related the real elapsed time between events to pattern similarity in the hippocampus and entorhinal cortex (Nielsen et al., *PNAS*, 2015; Bellmund et al., *eLife*, 2019; Thavabalasingam et al., *NeuroImage*, 2018; *PNAS*, 2019). It would thus not be implausible to assume that real time underlies the hippocampal representation in our task. Therefore, we believe our demonstration that the virtual times of events relate to hippocampal multi-voxel patterns even when accounting for variance explained by real time and order advances our understanding of how the hippocampus represents episodic sequences. In particular, this shows that, despite the evidence for the hippocampus tracking real time intervals and the necessity to track uncued real time intervals in our task, the hippocampal representation is subject to mnemonic construction.

Please find below the revised sections of the manuscript.

Figure 1 Caption

Thus, participants had to mentally construct event times by combining their experience of elapsing real time with the time cues.

Page 5

Participants had to combine their experience of objectively elapsing time (real time) with the virtual time cues to construct event times.

Page 13-14

In our paradigm, participants mentally constructed the times of events relative to a hidden virtual clock. To do so, they needed to combine their experience of passing real time with infrequent cues about the current virtual time. Thus, real time was critical for the successful construction of event times, despite not being cued explicitly. Participants' responses in a memory test and the similarity structure of hippocampal multi-voxel patterns were explained by virtual event times beyond the effects of real time and sequence order, showing that sequence representations reflect mnemonically constructed time. Recent work demonstrated the scaling of time cell representations to different real time intervals in the rodent hippocampus⁶⁷. Temporal scaling of hippocampal representations could potentially underlie our observation that temporal distances in virtual time are related to the similarity of event representations even when accounting for the effects of real time and order. This finding highlights that the anterior hippocampus maps relational knowledge derived from mnemonic constructions.

Page 18

Participants received no cues about elapsing real time, but had to use their experience of passing real time between virtual time cues to infer the event times relative to the hidden virtual clock.

Comment 4

From my prior review, not addressed: It is not clear why the authors chose to conduct two classes of analyses (summary statistics and mixed effects models) for every family of data. This is at best somewhat redundant, and at worst raises questions about correction for multiple comparisons when applying these analyses on the same family of data. Some reported effects fall between p-values of 0.025 and 0.05, meaning a simple Bonferroni correction would prove problematic. If both classes of tests are necessary, a clear case should be made for this approach. Otherwise, it may be prudent to choose one approach. If the authors believe that both classes of tests are necessary, and that multiple comparisons correction is unnecessary, this should be convincingly argued.

The reviewer here asks about the statistical analysis approaches we employed in our study. We decided to test our hypotheses using both a summary statistics approach as well as linear mixed effects models. The reason we opted for this is because both approaches offer distinct strengths. In the analysis of multi-voxel patterns, it is often recommended to employ non-parametric, permutation-based statistics (e.g. Stelzer et al., NeuroImage, 2013). These are readily incorporated in the summary statistics approach, where we run permutation-based procedures both on the subject and on the group level. Mixed models are becoming a more and more common tool in neuroscience and psychology (e.g. Yu et al., Neuron, 2021) and, because they do not discard within-subject variability, they have been argued to be more sensitive for effects in the underlying data. Further, they make it simple to test more complex interactions, such as the three-way interaction between regions of interest, virtual temporal distances, and sequence membership, which we report in the manuscript.

Importantly, we show that the results from the two analysis approaches converge. We do not believe that this is redundant, but rather that it demonstrates the robustness of our effects because they do not depend on the specific analysis that is employed, a strength also pointed out by Reviewer 2. Thus - while we understand the argument of the Referee - we decided to keep the two analyses approaches in the paper, but have now moved the presentation of the mixed model results to the supplements of the paper to make the key findings and figures more accessible, see point below. Yet, these analysis approaches cannot be viewed as independent because they test the same hypotheses on the same data. This is why we use an alpha-level of 0.05 for the two different ways of implementing those tests. We have made this explicit in the revised manuscript.

Please find the revised section of the manuscript below.

Page 24

We used a summary statistics approach, which uses permutation-based procedures on the subject-level as well as on the group-level, in line with recommendations for the analysis of multi-voxel patterns¹⁰⁵. We also implemented our statistical analyses using linear mixed effects models, which capture within-subject dependencies using random effects while estimating the fixed of interest on all data points. Mixed effects models are well-suited to test more complex interactions. The fact that the results of the two analysis approaches converge demonstrates that our findings are robust to the specific statistical technique. We used an α -level of 0.05 for both approaches because they are not independent as they are implemented on the same data and test the same hypotheses.

Minor Points:

Comment 5

Following from my final major point above, the figures are somewhat overwhelming, and carry seemingly redundant information. For example, in Figure 4 the authors present plots associated with a univariate regression and then follow with plots associated with a multiple regression. This depends on the authors' preferred solution to the above concern, but I will note that this figure and others would be simpler to parse if they only showed the multiple regression results, as they converge on the same conclusion as univariate regression, and as they are the stronger analysis via simultaneously evaluating multiple factors. I will note (in somewhat ironic contrast to the rest of this comment) that the authors have now added a multidimensional scaling analysis plot over the prior version of the manuscript, which while not adding anything terribly unique to the results, does provide a nice visualization of sequence representation.

The Reviewer here comments on the composition of our figures and the sequence of analyses presented for the within-sequence analyses of the hippocampus. We are pleased to learn that the Reviewer thinks that the multidimensional scaling results provide a helpful visualization of our results.

We acknowledge that the presentation of results from two parallel analysis streams resulted in multi-panel figures, which might have made it more challenging than necessary for the reader to understand our main findings. As discussed in response to the previous comment, we decided to keep the mixed model results in the manuscript. However, to make our figures less complex, we have moved the presentation of the mixed model results to the new Supplemental Figure 4. This figure now provides an overview of the mixed model results corresponding to the summary statistics analyses shown in the main figures of the manuscript. In our view, this change has made our figures more accessible for the reader. In response to a comment by a different Reviewer (Reviewer 1, Comment 1), we have moved the visualization of the raw pattern similarity change values for the same-sequence effect from Figure 5 to Figure 4 to better illustrate the direction of the effect.

With respect to the two analyses presented for the same-sequence effect in Figure 4, we believe that it is relevant to first show that virtual temporal distances correlate with hippocampal pattern similarity change before demonstrating that this effect holds also when including the other two time metrics in the multiple regression model. In terms of the progression of the manuscript, we think it is helpful to first show the effect of one time metric on pairs of events from one sequence before moving to the more complex multiple regression analysis. Also, we feel that it is necessary from a statistical point of view to demonstrate the effect of virtual time in isolation. If the multiple regression result was reported alone, there would be the possibility that the univariate regression effect would not be significant. Such a suppression effect could arise if real time and order would suppress variance of virtual time that is irrelevant to explaining pattern similarity change (Tzelgov & Henik,

Psychological Bulletin, 1991). For these reasons, we have decided to present both the univariate and the multiple regression results also in the revised manuscript.

Please find below the new Supplemental Figure 4, which shows the mixed model results corresponding to the summary statistics results shown in the main figures.

Supplemental Figure 4

Supplemental Figure 4. Mixed model results. Dot plots show parameter estimates and 95% confidence intervals for fixed effects of mixed model analyses. Line plots show estimated marginal means. **A, B.** Remembered times in the time line task are predicted by virtual event times with order and real time in the model (c.f. **Figure 2B**). **C, D.** Temporal distances in virtual time explain representational change in the anterior hippocampus (aHPC) for same-sequence events (c.f. **Figure 4B**). **E, F.** Temporal distances in virtual time explain representational change in the aHPC for same-sequence events when competing for variance with temporal distances based on order and real time (c.f. **Figure 4D**). **G, H.** Temporal distances in virtual

time explain representational change in the aHPC for different-sequence events (c.f. **Figure 5A**). **I, J**. There was a significant interaction of virtual temporal distances and sequence membership characterized by a differential relationship between temporal distances and aHPC representational change for event pairs from the same sequence or from different sequences (c.f. **Figure 5A**). **K, L**. Virtual temporal distances explain representational change in the anterior-lateral entorhinal cortex (alEC) when collapsing across all event pairs (c.f. **Figure 6B**). **M, N**. In the aHPC peak cluster of the same-sequence searchlight analysis, virtual temporal distances were significantly related to representational change for events from different sequences (c.f. **Figure 7B**). **O-R**. The relative time of events from other sequences predicted signed event time construction errors as measured in the timeline task (c.f. **Figure 8CD**) in the main fMRI sample (**O, P**) and in the independent replication sample (**Q, R**).

Comment 6

Throughout the text, the authors refer to sequences in temporal memory being “actively constructed.” Can claims about temporal relations being actively constructed be made based on pre vs. post task RSA? There is clearly some record of task structure in memory, but it is not clear that this is indicative of some active (rather than passive) process, or that the word “active” is really carrying any meaning here. This is a minor and somewhat nitpicky point, but given the repeated use of this phrase, it warrants some clarification or unpacking.

The Reviewer here comments on the use of the phrase “actively constructed” in relation to our RSA analyses, which are based on picture viewing tasks. In our view, inferring the virtual times of events requires an active constructive process as true virtual times are never shown. This construction likely takes place during the learning task and memory for the constructed times is precise in the post-scan memory test. We thus understand the concern about using this phrase when referring to data from the pre- and post-learning scan, where participants performed a target detection task that was matched between the pre- and post-learning scan. We have reworded the relevant sections of the manuscript accordingly to avoid confusion for the reader. Please find the changed sections below:

Page 2

However, whether event representations in the anterior hippocampus and anterior-lateral entorhinal cortex reflect temporal distances based on constructed event times is unclear.

Page 4

Here, we combine functional magnetic resonance imaging (fMRI) with a sequence learning task requiring the memory-based construction of the times of events forming different sequences.

Page 5

Participants had to combine their experience of objectively elapsing time (real time) with the virtual time cues to construct event times.

Page 5

With this paradigm, we partially dissociated the virtual time of events from the event order and real time to test whether mnemonically constructed event times underlie participants’ memory for the temporal structure of the sequences.

Page 8

Together, these data show that hippocampal representations of events from the same sequence changed to reflect mnemonically constructed event times.

Page 15

In conclusion, our findings show that the similarity of event representations in the hippocampus reflects relations between events that go back to mnemonically constructed event times, highlighting the impact of mnemonic construction on sequence memory beyond the effects of event order and real elapsing time.

Comment 7

The authors a-priori justification of their ROIs could stand to be fleshed out more. While I understand that prior work from this group has highlighted the anterior hippocampus and anterior-lateral entorhinal cortex, the logic for this selection warrants better justification. Especially given that many of the phenomena the authors allude to in the introduction and discussion are often associated with other regions.

We have followed the suggestion by the reviewer to describe in more detail why we chose the anterior hippocampus (aHPC) and the anterior-lateral entorhinal cortex (alEC) as our regions of interest. This choice is based on our previous work, where we showed that learning-induced pattern similarity changes in these regions reflect temporal distances between events in a sequence (Deuker et al., eLife, 2016; Bellmund et al., eLife, 2019). Further, these regions have been implicated in temporal memory and temporal coding more generally (for review: Bellmund et al., JoCN, 2020). Likewise, assessing cross-sequence generalization in the anterior hippocampus is supported by evidence and theoretical work suggesting that the hippocampus recombines episodic details across episodes (e.g. Preston et al., Hippocampus, 2004; Zeithamova et al., Neuron, 2021; Morton et al., PNAS, 2020; Whittington et al., Cell, 2020). We would like to note that our searchlight analyses, which test for the reported effects in our entire field of view, show that the strongest effects fall into the anterior hippocampus.

We have extended our description for why we chose the anterior hippocampus and the anterior-lateral entorhinal cortex as regions of interest for our analyses.

Page 7

We centered our analyses on the anterior hippocampus and the anterior-lateral entorhinal cortex (see Methods) based on our previous work implicating these regions in representing sequence relations^{21,27}.

Page 18

Our previous work demonstrates representations reflecting the temporal relations of events from one sequence in the anterior hippocampus²¹ and the anterior-lateral entorhinal cortex²⁷. More generally, these regions have been implicated in temporal coding and memory (for review, see¹⁰). Further, the hippocampus has been linked to inferential reasoning and generalization^{46,48,49,51,53}.

Comment 8

For Figure 8, why was the searchlight ROI used and not the anatomical ROI?

To assess the relationship between the behavioral generalization bias and sequence representations observed using RSA, we used the results of our searchlight analyses to quantify the strength of participants' hippocampal sequence representations. We based this analysis on the searchlight results because they provide a more precise localization of the effect in the hippocampus compared to the anatomically defined region of interest. We have clarified this in the revised manuscript.

Page 23

We chose this approach because the searchlight analyses provide greater spatial precision than anatomically defined region of interest masks.

REVIEWERS' COMMENTS

Reviewer #1 (Remarks to the Author):

I have reviewed the authors' response to my comments and those from the other 2 reviewers. I am satisfied with the revisions and do not have further suggestions. Overall, I believe the manuscript has been strengthened. Several of the new analyses help strengthen the claims (e.g., by more precisely isolating the role of virtual time) and the revised Discussion better addresses potential interpretations of the findings. All together, this is a very nice manuscript with rigorously analyzed data that will be of broad interest to readers.

Reviewer #2 (Remarks to the Author):

The authors are to be commended for considering each of my points carefully and conducting a number of additional analyses. The newly reported findings from these analyses have satisfactorily addressed my concerns and I have no further issues to raise. Well done to the authors for what is, in my opinion, an excellent piece of work that furthers our understanding of temporal memory and its representation in the MTL.

Reviewer #3 (Remarks to the Author):

The authors have done a fine job of addressing all of my questions and concerns. I commend them on a job well done, and have no further issues to raise.